# Measurements of nitric oxide and ammonia soil fluxes from a wet savanna ecosystem site in West Africa during the DACCIWA field campaign.

Federica Pacifico[1], Claire Delon[1], Corinne Jambert[1], Pierre Durand[1], Eleanor Morris[2], Mat J. Evans[2], Fabienne Lohou[1], Solène Derrien[1], Venance H. E. Donnou[3], Arnaud V. Houeto[3], Irene Reinares Martínez [1], Pierre-Etienne Brilouet[1]

[1]Laboratoire d'Aérologie, University of Toulouse, CNRS, UPS, Toulouse, 31400, France
[2]Wolfson Atmospheric Chemistry Laboratories, Department of Chemistry, University of York, York, YO10 5DD, UK
[3] Laboratoire de Physique du Rayonnement, Université d'Abomey-Calavi, Cotonou, 01 BP 526, Benin

*Correspondence to*: Claire Delon (claire.delon@aero.obs-mip.fr)

**Abstract.**

Biogenic fluxes from soil at a local and regional scale are crucial to study air pollution and climate. Here we present field measurements of soil fluxes of nitric oxide (NO) and ammonia ($NH_3$) observed over four different land cover types, i.e. bare soil, grassland, maize field and forest, at an inland rural site in Benin, West Africa, during the DACCIWA field campaign in June and July 2016. At the regional scale, urbanization and a massive growth in population in West Africa has been causing a strong increase in anthropogenic emissions. Anthropogenic pollutants are transported inland and northward from the mega cities located on the coast, where the reaction with biogenic emissions may lead to enhanced ozone production outside urban areas, as well as secondary organic aerosol formation, with detrimental effects on humans, animals, natural vegetation and crops. We observe NO fluxes up to 48.05 ngN $m^{-2}$ $s^{-1}$. NO fluxes averaged over all land cover types are 4.79 ± 5.59 ngN $m^{-2}$ $s^{-1}$, maximum soil emissions of NO are recorded over bare soil. $NH_3$ is dominated by deposition for all land cover types. $NH_3$ fluxes range between -6.59 and 4.96 ngN $m^{-2}$ $s^{-1}$. $NH_3$ fluxes averaged over all land cover types are -0.91 ± 1.27 ngN $m^{-2}$ $s^{-1}$ and maximum $NH_3$ deposition is measured over bare soil. The observations show high spatial variability even for the same soil type, same day and same meteorological conditions. We compare point daytime average measurements of NO emissions recorded during the field campaign with those simulated by GEOS-Chem (Goddard Earth Observing System Chemistry Model) for the same site and find good agreement. In an attempt to quantify NO emissions at the regional and national scale, we also provide a tentative estimate of total NO emissions for the entire country of Benin for the month of July using two distinct methods: upscaling point measurements and using the GEOS-Chem model. The two methods give similar results:

1.17 ± 0.6 GgN/month and 1.44 GgN/month, respectively. Total NH$_3$ deposition estimated by upscaling point measurements for the month of July is 0.21 GgN/month.

## 1 Introduction

Biogenic soil fluxes of nitric oxide (NO) and ammonia ($NH_3$) play an important role on tropospheric chemistry. Nitric oxide emitted by soil influences the concentration of nitrogen oxides ($NO_x$) in the atmosphere, consequently modifying the rates of ozone ($O_3$) production, where $O_3$ is a pollutant, harmful to humans and plants, and also a greenhouse gas (Steinkamp et al., 2009). The production and consumption of NO in soil is regulated by microbial activity, mainly nitrification/denitrification processes, and chemical reactions (Pilegaard et al., 2013). Measurements using soil chambers in the field and laboratory experiments show that nitrification/denitrification, and consequently NO emissions, vary greatly with climate and soil conditions, in particular they are strongly correlated with nitrogen (N) availability, temperature and soil moisture, making soil NO emissions dependent on regional temperature and precipitation patterns, and fertilizer management practices (e.g., Bouwman et al., 2002; Meixner and Yang, 2006; Hudman et al., 2010).

Soil NO emissions are about 20% of total NO sources to the atmosphere (IPCC, 2007) and almost of the same order of magnitude of fossil fuel NO emissions. Soil emission of biogenic NO plays a prominent role in the regional atmospheric chemistry of non-urbanized areas, where anthropogenic emissions are negligible (Pilegaard, 2013). The main inputs of N compounds onto semi arid uncultivated soils, like the savanna ecosystem, are biological nitrogen fixation, atmospheric wet and dry deposition and lightning. NO fluxes are considered as one way only, even if NO deposition exists in very specific conditions (Grote et al., 2009).

Soil N losses towards the atmosphere also involve $NH_3$. The largest sources of $NH_3$ emissions are animal husbandry, and agriculture via the application of synthetic fertilizer. When released into the atmosphere, $NH_3$ increases the level of air pollution. In the atmosphere $NH_3$ has a relatively short life time of less than five days and high deposition rates, it is converted into ammonium ($NH_4^+$) aerosols, which has a life time of the order of fifteen days, can travel long distances and it is relevant for air quality and climate (Fuzzi et al., 2015). The exchange of soil $NH_3$ is bi-directional as it also includes deposition. $NH_3$ returned to the surface by deposition can potentially cause eutrophication, reducing biodiversity and water quality (Sutton et al., 2009a).

The net flux of $NH_3$ is the combination of different exchange pathways between plant (cuticle and stomata), soil, leaf litter and atmosphere. The overall $NH_3$ flux for a given surface may switch from net emission to net deposition at sub-hourly, diurnal and seasonal scales. Moreover, $NH_3$ can be rapidly deposited onto cuticles due to its high solubility (e.g. Sutton et al., 2009b; Massad et al., 2010; Loubet et al., 2012).

The direction and magnitude of $NH_3$ exchanges depend on the difference in $NH_3$ concentration between the canopy and the atmosphere, and on a large range of environmental factors, in particular air humidity, which influence surface wetness, and soil moisture conditions, but also vegetation cover and soil characteristics. The relationships between NO and $NH_3$ soil fluxes have been identified through the ammonium content in the soil (McCalley and Sparks, 2008). Ammonia is mainly emitted by agricultural activities, and also by the decomposition of litter and volatilization of animal excreta (Sutton et al., 2009b; Massad et al., 2010).

Soil fluxes in West Africa have only been measured in a limited number of studies due to the challenging experimental conditions (remote sites, no power supply, very hot temperatures), and mainly with manual chamber techniques rather than more complex micrometeorological techniques (Serça et al., 1998, Le Roux et al. 1995 for NO, Delon et al., 2017 for NO and $NH_3$). However, tropical savanna has been recognized as one of the ecosystems characterized by the largest NO emissions (Davidson and Kingerlee, 1997, Hudman et al., 2012).

Anthropogenic emissions of pollutants from mega cities located on the Guinean coast in South West Africa have been increasing, and are likely to keep increasing in the next decades, due to a strong anthropogenic pressure, land use change and urbanization. When transported northward on the African continent, polluted air masses meet biogenic emissions from rural areas which contributes to increased $O_3$ and secondary organic aerosol production, in high temperature and solar radiation conditions, highly favorable to enhance photochemistry (Knippertz et al., 2015a, 2015b).

The objectives of this study are to quantify soil fluxes of NO and $NH_3$ for the different land cover types typical of rural West Africa, suggest a tentative strategy to scale point measurements in the field to ecosystem and larger regional scale, and provide data for inventories and model evaluation to improve air quality and climate modelling.

In this paper we present the soil fluxes of NO and $NH_3$ measured in a rural site near the city of Savè, Benin, West Africa, during the DACCIWA (Dynamics-Aerosol-Chemistry-Cloud Interactions in West Africa) field campaign which lasted from $14^{th}$ June to $30^{th}$ July 2016 (wet season). The DACCIWA campaign was lead to investigate the possible role of local air pollution on climate change in West Africa, focusing on atmospheric composition, air pollution and cloud–aerosol interactions over several sites in the region (Knippertz et al., 2015a, 2015b, 2017). The Savè site is part of the savanna ecosystem, where grassland is intercut with crops and degraded forest. Biogenic soil fluxes measurements were taken using the manual chamber technique, which is robust and of reduced costs (Delon et al., 2017). Along with these observations we also present measurements of soil characteristics and meteorological variables from the same site. We include the comparison of measured NO soil emissions with those simulated by the Hudman et al. (2012) process-based model for NO soil emission implemented into GEOS-Chem.

## 2 Material and method

 **2.1 Site description**

The Savè site for ground-based observations is located in a hinterland area of Benin, 6 km south west from the city of Savè (8°02'03" N, 2°29'11" E, 166 m a.s.l.). The Savè ground-based observation site is located within the Gobè site managed by the Institut National des Recherches Agricoles du Bénin (INRAB).

The site is characterized by a wet savanna ecosystem. The climate of the region is Sudano-Guinean, with a rainy season from March to October and a dry season from November to February (Michels et al., 2000). The average annual rainfall is about 1100 mm (Savè weather station, data averaged from 1969 to 2004, Michels et al., 2000 and Säidou et al., 2004) and the average yearly temperature is about 27.5 °C with little variation from year to year (data averaged from 1984 to 2004, Säidou et al., 2004). Average minimum temperature, based on 1969-1990 data, is 21.5 °C and mean maximum temperature is 35.5 °C.

The tree coverage in the Savè region is low with most of the land occupied by subsistence agriculture and grassland (CILSS, 2016). Four land cover types are identified at the observation site: bare soil, grassland, maize field and degraded forest. Bare soil is defined as a patch of land of minimum five by five meters wide, without vegetation growing or hanging over the plot. Ground photographs of the four land cover types are shown in Fig.1.

The most abundant tree species next to the grassland site and in the forest are: *Anacardium occidentale*, *Daniellia oliveri*, and *Pterocarpus erinaceus;* while the most abundant tree species next to the maize field are: *Mangifera indica*, *Cocos nucifera*, *Carica papaya L.*, *Tectona grandis*, and *Azadirachta indica*. The herbaceous vegetation is dominated by *Cleome* sp., *Crotalaria* sp., *Mucuna* sp., *Imperata cylindrica* and *Rhynchelytrum repens* next to the grassland site and in the forest, and *Commelina benghalensis*, *Euphorbia* sp., *Boerhavia diffusa*, *Phyllanthus amarus*, *Digitaria horizontalis* by maize field. In the maize field, the main species, *Zea Mays*, is intercropped with *Sesamum indicum* and, to a lesser extent, with other species: *Dioscorea* sp., *Manihot esculenta*, *Arachis hypogaea*, *Vigna unguiculata*, *Gossypium* sp., *Sorghum* sp. and *Solanum lycopersicum*. The maize field was not treated with mineral fertilizer. The only livestock consist of a few dozens of domestic fowls belonging to small subsistence-oriented family farms, mainly grazing in the maize field.

At the Savè site, the soil is sandy, with 87% of sand and 4.1% of clay (the rest being silt) for the 0-5 cm horizon. Surface pH ranges from 6.32 to 8.46, depending on the place where the measurement is done. Mean meteorological and average soil characteristics for the observation site are reported in Table 1, dominant vegetation species and soil composition for each land cover type are given in Table 2 and 3, respectively. Sunrise and sunset UTC time at the beginning and at the end of the campaign are: 05:33 and 18:08 on 14th June, and 05:42 and 18:11 on 30th July.

## 2.2 Sampling sites

The samples were taken at the four land cover types (bare soil, grassland, maize field and forest), one location per day. Two to three samplings spots were chosen each day for each location, collecting eight to twenty-five flux measurements for both NO and $NH_3$ soil fluxes each day. Each location was sampled during daytime, approximately from 7 a.m. to 6 p.m., alternating measurements at the four different land cover types from one day to the other, over the entire campaign. Bare soil and the maize field were sampled for both NO and $NH_3$ soil fluxes on eight different, generally non-consecutive, days, grassland on ten days, and the forest site on four different days.

## 2.3 Chamber flux measurements

The technique used to measure NO and $NH_3$ soil fluxes makes use of a Thermoscientific 17i (ThermoFischer Scientific, MA, USA). This analyzer uses a chemiluminescence detector for NO. The air sample enters the reaction chamber and reacts with the $O_3$ generated by an internal generator. This reaction produces a luminescent radiation directly proportional to the NO concentration. The air sample is sequentially drawn through a molybdenum converter heated to 325°C which measures NOx ($NO+NO_2$) by converting $NO_2$ to NO, and a stainless steel converter heated to 750°Cwhich measures Ntotal ($NH_3$+NOx) by converting $NH_3$ and $NO_2$ to NO. The detector hence measures rNO, then r(NO + $\alpha NO_2$), and finally r(NO + $\beta NO_2$ + $\gamma NH_3$), where r is the NO detection efficiency, $\alpha$ and $\beta$ are the $NO_2$ conversion efficiency of the molybdenum and stainless steel converters and $\gamma$ is the $NH_3$ conversion efficiency of the stainless steel converter. The efficiencies are determined by the calibration procedure. $NH_3$ concentration is therefore calculated from Ntotal – NOx. The closed dynamic chamber technique is used to calculate fluxes. The details of this technique are fully described in Delon et al. (2017).

The remoteness of the study site limited installation of permanent structures and we were unable to automate our chamber measurements, thus all measurements were made manually. The instrument was powered by a generator (>100 m away) and carried around on a wheeled-table to reach the locations of the four soil types where the NO and $NH_3$ soil fluxes were measured. The analyzer was connected via a Teflon tube to the Teflon chamber which was put on the ground to detect the fluxes. The external sides of the chamber were covered with sand or soil to isolate it during the measurement. The soil under the chamber was left unperturbed. Adjustments were taken in order to make sure the analyzer did not reach temperatures that would invalidate the measurements.

The calibration of the NO sensor of the 17i analyzer was made before and after the campaign, with a reference NO air mixture, i.e. NO in $N_2$ diluted with zero air. The NO detection efficiency variation was 8% between the two calibrations (from 1.040 to 0.962). Two post-campaign calibrations were made: a first one to validate the efficiency of the $NO_2$ converter using a reference dilution of $NO_2$ in zero air, and a second one to validate the efficiency of the $NH_3$ converter with a $NH_3$/$N_2$

mixture diluted in pure air (Alphagaz 1, Airliquide). No change in the $NO_2$ conversion efficiency was necessary, and the $NH_3$ conversion efficiency variation was 3% (from 0.963 to 0.995). No drift in the conversion efficiencies was observed over time, as from the first calibration when the analyzer was new until the post campaign calibration, changes never exceeded ±3%. The zero air for NO, $NO_2$ calibration was obtained by filtering ambient air, previously passed on charcoal and desiccant cartridges. The dilution for all the calibration experiments was made with the 146i module (ThermoFischer Scientific, MA, USA) and the dilution module, equipped with certified mass flow meters, on board of the ATR-42 research aircraft during an inter-calibration with other $NO_x$ instrumentation of the DACCIWA campaign (i.e. the instrumentation on the Savè measurement site tower and the instrumentation on the ATR-42 aircraft; Brito et al., 2017, Derrien et al., 2016). Reference NO, and $NO_2$ were ISO 6141:2015 certified at 8.73 and 8.58 ppm for NO, before and after the campaign, respectively, and 9.28 ppm for $NO_2$, both with 5% precision. Reference $NH_3$ mixture was certified at 14.78 ppm with 2% precision for $NH_3$. Multipoint (at least 4 points) calibrations between 50 to 250 ppb were done to ensure the linearity of the response, obtaining regression coefficients over 0.9993 for both NO and $NO_2$. The dilution uncertainty was 10% for NO, 11% for $NO_2$ and 13% for $NH_3$ (see Appendix A for more detail). A multipoint calibration was done for $NH_3$, between 30 and 200 ppb and the regression coefficient was 0.997. The linearity of the response for low concentrations is tested by the response to zero air calibration, giving a $R^2$=0.997. However, at low mixing ratios (typically less than 100 ppb), a non linear increase of the interactions of $NH_3$ with the surface used in the inlet design has to be considered (Ellis et al., 2010, Whitehead et al., 2008). Therefore, an uncertainty in the quantification of low $NH_3$ concentrations has to be taken into account due to surface interactions. The global precision of the analyzer is ±0.4 ppb according to the manufacturer's specification for a 0-500 ppbv range.

The external volume of the chamber was 40 cm×20 cm×20 cm. The internal volume was $18×38×18$ cm$^3$, due to the thickness of the Teflon walls. The air inlet is located on one side of the chamber, where a small vent of 4 mm in diameter provided the pressure equilibrium between the inside and outside of the chamber. The air outlet on the other side is connected to the analyzer with a 4 m Teflon tube (see picture displayed in Appendix B) The chamber is continuously swept with an air flow of 0.7 L min$^{-1}$ insured by the instrument pump, and the air flow is controlled inside the analyzer by a flow meter. The air residence time in the chamber is approximately 20 min (Volume/flow), and the chamber is maintained in place for 10 min. The Teflon chamber was cleaned (with a dry clean paper cloth) at the beginning of each day of measurement, and during the day when the deposition of sand could potentially interfere with the measurements. Laboratory tests using different papers for cleaning are displayed in Appendix C. According to these results, no clear tendency for potential adsorption or desorption of $NH_3$ arises but these tests may be useful to warn for a potential pollution inside the chamber due to cleaning which would interfere with low fluxes.

The opaque walls minimize photochemical reactions inside the chamber, which are therefore considered as negligible. The chamber is placed on the soil for 10 min. After 10 min, the chamber is turned over to let the analyzer be swept by ambient air for 5 min, then the chamber is placed again on the soil to begin a new cycle.

The calculation of the fluxes is based on the closed dynamic chamber technique, with the following assumptions: the concentration in the chamber is equal to the concentration leaving the chamber to the analyzer, and potential deposition onto the Teflon walls of the chamber is assessed but considered as negligible. Vaittinen et al. (2013, and references therein) have demonstrated that the adsorption of ammonia on Teflon is negligible; However, the high $NH_3$ mixing ratios and the controlled conditions in Vaittinen experiment do not correspond to our field conditions. Therefore, experimental tests with and without the Teflon chamber attached to the analyzer were made in ambient air to verify that deposition on the walls of the Teflon chamber is negligible. These tests have been made in conditions comparable to in situ measurements, i.e. temperature (25 to 29°C) and humidity (46 to 54%), as well as $NH_3$ concentrations (8 to 35 ppb) close to the ones encountered in the field. They show that the concentrations measured with and without the chamber are equivalent. . The results of this experiment are reported in Appendix B. Moreover, the temperatures of the chamber Teflon walls and Teflon tube have been measured in direct sunlight and the difference with air temperature is small (<1°C). We therefore assume that the Teflon walls and tube heating is small and does not affect the $NH_3$ and NO concentration measurements in the chamber. The results of this experiment are reported in Appendix D. All the details of the calculation are given in Delon et al. (2017). In brief:

$$F_x = \frac{V}{A_0} \frac{\delta C_x}{\delta t} \qquad (1)$$

Where $F_X$ is the flux (NO or $NH_3$) in nmol m$^{-2}$ s$^{-1}$, $\delta C_X$ is the concentration variation in the chamber in nmol m$^{-3}$ during the temporal interval $\delta t$. $A_0$=0.0684 m$^2$ is the surface of the ground covered by the chamber, V=0.0123 m$^3$ is the volume of the chamber. This equation is similar to the one in Davidson et al. (1991). The flux is then converted to ngN m$^{-2}$ s$^{-1}$.

The linear regression is calculated over a 100 to 300 s time interval after the installation of the chamber on soil for both NO and $NH_3$. The dilution effect due to mixing of outside air in the chamber was evaluated based on our set up in which Q/V=8.13×10$^{-4}$ s$^{-1}$. It is calculated for each flux separately and is in average 6.7(±1.6) % for NO and 7.7(±1.7) % for $NH_3$. Considering the precision of the analyzer (±0.4 ppbv), the detection limit is 0.4 ngN m$^{-2}$ s$^{-1}$ for NO and $NH_3$ fluxes. According to Appendix B, if the difference in $NH_3$ concentration used to calculate a flux is inferior to 0.9 ppb, the resulting flux may not be distinguished from a potential effect of adsorption or desorption onto the chamber walls. The precision of the analyzing device (analyzer + chamber + tube) may be defined at 0.9 ppb (corresponding to a flux of 0.55 ngN.m$^{-2}$.s$^{-1}$). In that case, low $NH_3$ fluxes comprised between 0.4 and 0.55 ngN.m$^{-2}$.s$^{-1}$ are considered as close to zero but are kept in the average daily flux calculation.

The chemical reactions inside the chamber can determine NO consumption, and consequently an underestimation of the NO fluxes calculated with our method. This underestimation is taken into account and calculated following the method by Pape

et al. (2009) with the relation k·[NO]·[O$_3$]. In this relation k is the temperature-dependent reaction rate constant (Pape et al., 2009, Atkinson et al., 2004), [NO] is measured by the Thermoscientific 17i at soil level just before positioning the chamber for the measurement of soil fluxes, and [O$_3$] at soil level is derived by measurements of NO and NO$_2$ at soil level made with the Thermoscientific 17i and measurements of NO, NO$_2$ and O$_3$ taken on an 8 m high tower. On the 8 m high tower, NO and NO$_2$ were measured with a Model 42C TraceLevel NO-NO$_2$-NOx by Thermo-environmental Instruments Inc., calibrated with the same method as the Thermoscientific 17i and with 0.05 ppb (2-sigma) detection limit. Ozone was measured on the tower with a Model 49i Ozone Analyzer by Thermo-environmental Instruments Inc. with 1 ppb detection limit. The Model 49i Ozone Analyzer was calibrated by comparison with a Thermo Scientific Model 49PS reference instrument. The reference instrument is sent twice a year to the French Laboratoire national d'Essais (LNE) for comparison with a National Institute of Standards and technology (NIST). All data on the tower were sampled at 10 seconds. [O$_3$] at soil level was then calculated considering the diurnal steady state of the reactions described in equation (2) and (3), using equation (4):

$$NO + O_3 \rightarrow NO_2 + O_2 \tag{2}$$

$$NO_2 \rightarrow NO + O \tag{3}$$

$$[O_3]_{sl} = \frac{[NO]_{tl}[O_3]_{tl}}{[NO_2]_{tl}}\frac{[NO_2]_{sl}}{[NO]_{sl}} \tag{4}$$

Where []$_{sl}$ is the concentration at the soil level and []$_{tl}$ is the concentration measured on the tower. In conclusion, we correct NO fluxes for the underestimation of NO fluxes due to chemical reactions inside the chamber with values ranging between 0 and 63% (8% on average for the whole campaign).

As studied by Kristensen et al. (2010a) and Kristensen et al. (2010b), O$_3$ deposition can decrease O$_3$ concentration close to soil surface further. However, considering that O$_3$ concentrations calculated near the soil are already very low (1 ppb at soil level compared to 24 ppb at 8 m, averaged for the entire measurement campaign), O$_3$ deposition has been considered of secondary importance in this calculation and has not been included. If O$_3$ deposition were to be included it would possibly decrease the correction of NO fluxes and consequently slightly decrease NO emissions in a negligible proportion compared to the correction already applied for the chemical reactions inside the chamber.

NH$_3$ measurements have not been corrected from a possible interaction with particulate matter (PM) as PM concentration (not measured at Savé) are supposed to be low because Savé is located in a rural area far from anthropogenic pollution influence. The walls of the Teflon chamber are cleaned daily to reduce any interference of NH$_3$ with PM deposition in the chamber. The effect of PM, even at low PM concentrations, may reduce the measurement accuracy and induce an uncertainty on the detection of the NH$_3$ flux from soil. This uncertainty has not been assessed quantitatively, but the reader must keep in mind that NH$_3$ fluxes may be estimated with less accuracy because of the presence of PM, especially for low fluxes.

**2.4 Data quality check**

A quality check method based on the following criteria is used to select observed fluxes (Delon et al., 2017):

- The coefficient of determination for linear regression $R^2$ has to be higher than 0.4 (considered as a significant correlation) for $NH_3$ fluxes, and higher than 0.8 for NO fluxes. The variation of $NH_3$ is less stable than for NO because of potential interaction with PM in the chamber. However, 80% of the $R^2$ were superior to 0.6 for $NH_3$, and 100% for NO. Examples of the variation in time of the concentration of $NH_3$ and NO in the chamber are shown in Appendix E for two different soils.

- A flux error was estimated by calculating the dispersion of points around the linear regression's slope. According to this method, the dispersion for NO flux calculation is comprised between 5 and 12%, and the dispersion for $NH_3$ flux calculation is comprised between 15 and 20%.

- The concentration difference between the last and the first $NH_3$ measurement point has to be more than 0.4 ppb (sensitivity of the analyzer). $R^2$ was generally lower than 0.4 for concentration differences below 0.4 ppb.

Finally, 351/488 (72%) $NH_3$ flux measurements and 459/488 (94%) NO flux measurements are considered valid. Among the 351 $NH_3$ valid fluxes, 30% are derived from a concentration difference of less than 0.9 ppb.

Uncertainty of the $NH_3$ flux calculation

Despite all precautions to reduce adsorption on the chamber walls and/or interaction with PM in air, (see Appendix B, C D
and E), $R^2$ are less good for $NH_3$, due to potential chemical or physical interaction of material with $NH_3$ (whereas considered as negligible in this study). However, no absolute correction for adsorption can be calculated in field conditions. Teflon remains the more reliable material to measure $NH_3$, as shown in Sauren et al. (1989) who find that Teflon has the lowest adsorption affinity for $NH_3$ (as compared with aluminium, parafine and gold), but a passivation time lag remains for $NH_3$ detection in measurements systems (Yokelson et al., 2003).

**2.5 Meteorological station**

Continuous in situ observations of meteorological variables, including air and soil temperature and moisture, rainfall, wind speed, wind direction, radiation and energy balance components were taken at the Savè site as part of the DACCIWA campaign. Data are provided as 1 min averages, apart from energy fluxes which are given as 30 min averages (Derrien et al., 2016, Kohler et al., 2016, Handwerker et al., 2016, Wieser et al., 2016). An overview of the complete set of instrumentation
and measurements is given by Brooks et al. (2017), while a summary of the available ground-based meteorological observations is given by Kalthoff et al. (2017). In this study we present soil moisture measured in two distinct locations of the Savè site by the Karlsruhe Institute of Technology (KIT) instrumentation at 5cm depth on grassland, and average soil moisture, between 0 and 30 cm, measured by the Université Paul Sabatier (UPS) instrumentation in the maize field. Details of the instrumentation is given by Brooks et al. (2017). We include soil moisture measured with both systems as the inter-
comparison of the two methods is out of the scope of this study.

**2.6 Soil characteristics (texture, pH, N content)**

Soil samples were collected with a cylinder of known volume (290 cm$^3$) during the measurement campaign to analyze the biogeochemical characteristics of the site. Soil samples (0-5 cm) were taken for each land cover type where NO and NH$_3$ fluxes were measured. Fifteen samples were collected at the four different land cover types, three to four times during the campaign.

Samples were dried in ambient conditions (mean day-time temperature is approximately 26 °C, Kalthoff et al., 2017), and stored in the dark. After drying, the weight of the samples was measured to determine the bulk density (d$_a$ = dry soil mass / total volume), which was found to be 1.24 ±0.14g cm$^{-3}$ Assuming a density of soil particles (d$_r$) of 2.6 g cm$^{-3}$ , the Water Filled Pore Space (WFPS) is calculated with the following equation 5:

WFPS = SM/(1-d$_a$/d$_r$)          .                                                                                (5)

where SM is soil moisture in %

Soil samples were analyzed for the determination of texture, ammonium concentrations [NH$_4^+$], C/N ratio, total C, total N, and pH at the GALYS Laboratoire (http://www.galys-laboratoire.fr, NF EN ISO/CEI 17025: 2005). The analyses were performed two months after sampling. We assume that the ammonium content in litter or soils is not modified by volatilization or chemical transformation during transport and storage, because of the very low soil moisture level in samples. Indeed, when collected, WFPS of the samples ranged between 6 and 14% (mean=8.5±3.5%) and soil temperature between 35 and 38 °C (data obtained from the databases described in Brooks et al., 2017). Bai et al. (2013, and references therein) have found that significant changes in nitrification and net mineralization (influencing the ammonium content) may occur when soil temperature raises until 35°C (the optimum for nitrification), for optimal soil moisture conditions ( WFPS =20%, Oswald et al., 2013). In the present study, soil temperatures when sampling were equal or above the optimum, and WFPS was below the optimum, reducing the nitrification efficiency and the change in ammonium content. Several authors have published results of ammonium concentrations measured in soils dried in ambient air. For example, Dick et al., (2006) collected top soil after the wet season in two sites in Senegal. The authors state that their soils were considered dry when collected and were air-dried in the mid-day sun immediately after collection. The protocol used in our study is identical. Other studies (Bai et al., 2010, Cassity-Duffrey et al., 2015, Vanlauwe et al., 2002) also published ammonium measurements made on air dried soils from seasonally dry climates with comparable textures to the soil in Savé. Soil texture is determined following norm NF X 31e107. Clay (<2 μm), fine silt (2 to 20 μm), coarse silt (20 to 50 μm) and total sand (50 to 2000 μm) are determined without decarbonation. Organic carbon and total carbon are determined following norm NF ISO 10694. The whole carbon of the sample is transformed into $CO_2$. Then $CO_2$ is measured by thermal conductibility. NF ISO 13878 is used for Total N. Mineral nitrogen is determined following an internal method MT-AZM adapted from norm NF ISO 14256-2. This method uses a potassium chloride solution and is COFRAC certified. The sample is heated at 1000°C with $O_2$. Products

of combustion or decomposition are reduced in $N_2$. $N_2$ is then measured by thermal conductibility (catharometer). pH is determine!d according to norm NF ISO 10390, with soil samples stirred with water (ratio 1/5).

## 2.7 Soil ammonia emission potential $\Gamma_g$ and compensation point $\chi_g$

Measurements of soil pH and ammonium concentrations $[NH_4^+]$ are used to quantify the soil emission potentials for the different land cover types at the measurement site. The soil emission potential $\Gamma_g$ is the ratio of $[NH_4^+]$ to $[H^+]$ concentrations in the water solution of the soil (mol $L^{-1}$). A large $\Gamma_g$ indicates that the soil has a high propensity to emit $NH_3$, considering that the potential emission of $NH_3$ depends on the availability of ammonium in the soil and on pH.

The soil compensation point ($\chi_g$) has been calculated from the emission potential $\Gamma_g$, as a function of soil surface temperature ($T_g$ in K) according to Wentworth et al. (2014):

$$\chi_g \text{ (ppb)} = 13\,587 \cdot \Gamma_g \cdot e^{-(10\,396K\,/\,Tg)} \times 10^9, \qquad\qquad (6)$$

The soil compensation point indicates the equilibrium between gaseous $NH_3$ in the soil pore space and $[NH_4^+]$ in the soil solution, i.e. the concentration of $NH_3$ for which the $NH_3$ flux switches from emission to deposition (or vice versa).

## 2.8 Stepwise multiple regression analysis

A stepwise linear multiple regression analysis was performed between daytime averaged gas fluxes of NO and $NH_3$ and relevant available daily averaged variables such as wind speed, soil temperature at 5 cm, soil moisture at 5 cm, soil heat flux, outgoing longwave radiation and incoming shortwave radiation. Soil parameters such as mineral nitrogen, total N and organic C, soil texture and pH could not be used for this regression analysis since their relative measurements did not have the same temporal resolution as the other parameters. The R software (http://www.R-project.org) was used to provide the results of this linear regression analysis.

## 2.9 GEOS-Chem

GEOS-Chem is a global three-dimensional model of tropospheric chemistry driven by meteorological input from the NASA Goddard Earth Observing System (www.geos-chem.org, Bey et al., 2001). In this study we use GEOS-Chem Version 10-01 which includes the process-based parameterization of soil NO emission by Hudman et al. (2012). This parameterization represents available nitrogen (N) in soils using biome specific emission factors, online wet and dry deposition of N, and fertilizer and manure N derived from a spatially explicit dataset, distributed using seasonality derived from data obtained by the Moderate Resolution Imaging Spectrometer (MODIS). Emissions are a smooth function of soil moisture and temperature consistent with point measurements and ecosystem scale experiments. This parameterization also included pulsing following

soil wetting by rain or irrigation, represented as a function dependent on dry spell length. The parameterization by Hudman et al. (2012) was successfully evaluated for pulsing events in central Sahel (0–30° W, 12–18° N).

Boundary conditions for our experiment are generated from a global GEOS-Chem simulation at 4° x 5° horizontal resolution. The regional GEOS-Chem model for West Africa runs at a horizontal resolution of 0.25° x 0.3125° (latitudes 6°S–16°N, longitudes 18.125°W–26.875°E) and a vertical resolution of 47 levels (up to 0.01hPa). Meteorology is driven by the NASA GMAO (Global Modeling and Assimilation Office) GEOS-FP (Forward Processing) assimilated meteorological data. The global model is spun-up from 1st May 2015 to 1st May 2016. The global simulation is then run from 1st May 2016 to 1st August 2016, outputting boundary condition files for West Africa. The regional West Africa simulation is then run from 1st May 2016 to 1st August 2016 using the 4° x 5° boundary conditions from the global simulation. All simulations use the GEOS-FP meteorology which has a three-hour time resolution. We used the same MODIS/Koppen land cover map as in Hudman et al. (2012; http://glcf.umd.edu/data/lc) which includes 24 land cover types. In this simulation we use EDGAR v4.2 (EC-JRC/PBL, 2011) for anthropogenic emissions, GFED4 (Giglio et al., 2013) for biomass burning emissions and MEGAN v2.1 (Guenther et al., 2012) for biogenic emissions of volatile organic compounds. The same emission inventories are used for both the boundary conditions and the West Africa simulation.

## 3 Results and discussion

### 3.1 Meteorological data

Mean air temperature averaged over the whole campaign was 25.4 ± 2.6 °C, mean wind speed was 1.3 ± 0.6 m s$^{-1}$, mean relative air humidity is 86.3 ± 10.5 %, mean soil temperature was 25.2 ± 3.4 °C, mean KIT soil moisture at 5 cm was 7.1 ± 3.6 %, while mean UPS soil moisture averaged between 0 and 30 cm was 4.5 ± 2.8 %. Total KIT precipitation was 198 mm for the whole campaign, and total UPS precipitation was 215 mm.

Median diurnal cycles of air temperature, specific humidity and precipitation are reported in Kalthoff et al., (2017). Knippertz et al. (2017) distinguish four different phases of the monsoon season during the DACCIWA campaign (14th June to 30th July 2016) over the DACCIWA focus region (5-10° N, 8° W - 8° E), which covers a wide area of West Africa (see Fig. 1, Knippertz et al. (2017)). The division into phases is mainly based on the north–south precipitation difference between the coastal zone (0–7.5° N) and the Sudanian–Sahelian zone (7.5–15° N), both averaged across the longitude range 8° W–8° E. Savè (8.03° N) is located very close to the border between the two zones, with a rainfall pattern that seems to follow more closely that of the coastal zone rather than that of the northern inland Sudanian–Sahelian zone. These four phases are: the pre-onset phase characterized by a rainfall maximum near the coast (before 21st June, phase 1); the post-onset phase during which the rainfall maximum occurred inland (22nd June - 20th July, phase 2); the wet westerly regime when the rainfall maximum shifted back to the coast (21st – 26th July, phase 3); and the recovery of the monsoon with a shift of the rainfall

maximum inland (27[th] July until the end of the campaign, phase 4). A specific period within phase 2 is indicated "vortex", during which an unusual development occurred (09[th] – 16[th] July): in the north, a cyclonic feature slowly propagated from eastern Mali to Cape Verde and in the south, an anticyclonic vortex tracked in the west-northwesterly direction along the Guinean coast (see Knippertz et al. 2017 for a more detailed description). At the Savè site the most intense rainfall events happened the day before the first soil fluxes observation, on 15[th] June 2016, and towards the end of the measurement campaign between 20[th] and 23[rd] July 2016. Other minor rainfall events are recorded on 19[th] and 27[th] June, 8[th], 12[th], 13[th], 24[th] and 26[th] July. Daily rainfall measurements are reported in Figs. 2 to 5.

## 3.2 Soil texture, soil organic carbon, total nitrogen, pH and ammonium content

Bare soil recorded the lower amount of total sand (83.75 ± 1.82 %) and the higher amount of clay (5.13 ± 0.63 %), fine (5.13 ± 0.96%) and coarse silt (5.98 ± 0.51%). Grassland recorded the higher amount of total sand (89.20 ± 0.71 %) and the lower amount of clay (3.15 ± 0.50%) and fine silt (2.93 ± 0.32%), while intermediate values were found for the maize field and forest (Table 3). These values determine the classification of sandy soil for all measurements sites.

Soil organic carbon (C) and total nitrogen (N) are respectively 12.2 ± 5.7 g kg[-1] and 0.95 ± 0.51 g kg[-1], averaged for all land cover types over the entire campaign. Table 4 gives soil characteristics for each land cover types, including individual values of C/N ratio, soil organic C and total N for the entire field campaign. The highest average soil organic C was measured for bare soil (17.3 ± 5.9 g kg[-1]) and the lowest soil organic C was measured for grassland (6.2 ± 1.3 g kg[-1]), while the maize field and forest site accounted for 14.1 ± 2.9 g kg[-1] and 11.4 ± 4.5 g kg[-1] soil organic C, respectively. The highest average total N was measured for bare soil (1.44 ± 0.51 g kg[-1]) and the lowest total N was measured for grassland (0.44 ± 0.04 g kg[-1]), while the maize field and forest site accounted similar amounts of total N, 0.99 ± 0.19 g kg[-1] and 0.94 ± 0.48 g kg[-1], respectively. Values of C/N, soil organic C and total N recorded for grassland at the Savè site compare closely to those reported by Delon et al. (2017, table 2) for the semi-arid site of Dahra (15°24' N 15°25' W), Senegal. Our values of C/N and total N for grassland are also close to those reported by Le Roux et al. (1995, table 1) and Lata et al. (2004) for the wet savanna ecosystem of Lamto (6°13' N, 5°20' W), Ivory Coast, although we observe lower values of soil organic C compared to these studies. Values of C/N and soil organic C recorded for the maize field at the Savè site are slightly higher than those recorded by Barthes et al. (2004) in a maize field at Agonkanmey (6°24' N, 2°20' E), near Cotonou in southern Benin.

All the sites listed in the comparison in the previous paragraph are sandy, as the Savè site. The Dahra site (Delon et al., 2017) also shows similar pH than our site (Table 5), while lower pH (acidic or near-neutral) was recorded at the sites of Lamto (Le Roux et al., 1995, Lata et al., 2004) and Agonkanmey (Barthes et al., 2004). Table 5 provides individual values of pH, [$NH_4^+$], $\Gamma g$ and $\chi_g$ for the entire field campaign. The highest average pH was observed for bare soil (8.23) and the lowest for the forest site (7.07), while measured average pH was 7.27 for grassland and 7.70 for the maize field. The [$NH_4^+$] content averaged for all land cover types over the entire campaign is 5.33 ± 4 mg kg[-1]. The highest average [$NH_4^+$] was recorded for

the maize field (7.9 ± 6 mg kg$^{-1}$) and the lowest for grassland (2.0 ± 0.3 mg kg$^{-1}$). Average [NH$_4^+$] is 6.2 ± 5 mg kg$^{-1}$ and 7.0 ± 2.2 mg kg$^{-1}$ for forest and bare soil, respectively. Dick et al. (2006) have found NH$_4^+$ concentrations between 2 and 8 mgN.kg$^{-1}$ in Senegalese soils, which is very close from our results. Vanlauwe et al. (2002) have found values between 0.8 and 1.4 mgN.kg$^{-1}$ in West African moist savanna soils (in Togo and Nigeria).

Higher soil organic C and N over bare soil could be due to the fact that these bare soil patches experienced recent burning (Santín and Doerr, 2016). The higher [NH$_4^+$] over the maize field can be caused by chicken excreta, as chickens were roaming over the maize field (Paillat et al., 2005, Tiquia and Tam, 2000).

### 3.3 Soil emission potential $\Gamma_g$ and compensation point $\chi_g$

The mean soil emission potentials for the Savè site is 43 714 ± 58 077, with values ranging from 380 to 159 343. The highest values of soil emission potentials are observed for bare soil (113 672 ± 67 788), followed by maize field (33 880 ± 20 680), forest (11 982 ± 11 061) and grassland (4 929 ± 4 409). The ammonia compensation point ranges between 5 to 2 215 ppb, with soil temperatures between 25 and 29 °C. The highest values of $\chi_g$ are observed for bare soil (1 607 ± 993), followed by maize field (473 ± 317), forest (175 ± 167) and grassland (58 ± 47). Our values of soil emission potential for bare soil and maize (no fertilization) are comparable with those presented in Massad et al. (2010, table 4), although those data come from measurements taken on different ecosystems. Both $\Gamma_g$ and $\chi_g$ values recorded at the Savè site exceed those recorded by Delon et al. (2017) over a grazed semi-arid Sahelian ecosystem in Senegal.

### 3.4 NO fluxes

NO fluxes from soil measured during the field campaign range between 0 and 48.05 ngN m$^{-2}$ s$^{-1}$. NO fluxes averaged over all land cover types are 4.79 ± 5.59 ngN m$^{-2}$ s$^{-1}$, while average NO fluxes for each land cover type are: 8.05 ± 3.49 ngN m$^{-2}$ s$^{-1}$ for bare soil, 3.73 ± 1.76 ngN m$^{-2}$ s$^{-1}$ for the maize field, 2.87 ± 1.49 ngN m$^{-2}$ s$^{-1}$ for forest and 2.82 ± 3.46 ngN m$^{-2}$ s$^{-1}$ for grassland. Soil emissions of NO from the different land cover types provide similar values, NO emissions from bare soil are higher on average, but have a larger standard deviation (Table 6).

Other measurements of biogenic NO soil emissions from the West African wet savanna can be found in Delon et al. (2012, table 7). We find that our measured NO soil emissions averaged over all land cover types are higher than those measured from other wet savanna sites. Our measurements are in better agreement with emissions from dry savanna grasslands (Delon et al., 2012), and with measurements from a semi-arid savanna, with over 80% sandy soil, in South Africa (Parsons et al., 1996, Scholes et al., 1997). However, these studies measured NO emissions during different seasons and soil moisture conditions compared to our study. For example, Parsons et al. (1996) recorded NO emissions up to 20 ngN m$^{-2}$ s$^{-1}$ over an open savanna during the period going from the end of the dry season to the beginning of the wet season. Nitric oxide

emissions of the same magnitude as in our study were also recorded over a grazed semi-arid Sahelian ecosystem in Senegal during the month of July by Delon et al. (2017): $5.7 \pm 3.1$ ngN m$^{-2}$ s$^{-1}$ in July 2012 and $5.1 \pm 2.1$ ngN m$^{-2}$ s$^{-1}$ in July 2013.

Daytime means of NO concentrations are measured close to the soil (0.1m, half height of the chamber) and reported in fig. 2 to 5. Daytime means of NO concentration vary from 1.28 to 5.40 ppb for all sites. The average concentration during the whole campaign on all sites is $2.70 \pm 1.03$ ppb. Average NO concentration is $2.97 \pm 1.49$ ppb on bare soil, $2.57 \pm 0.96$ ppb on grassland, $2.55 \pm 0.83$ on maize, and $2.76 \pm 0.65$ ppb on forest soil (Table6). The concentrations are quasi equivalent for all sites. As these concentrations are low, they do not lead to NO deposition on soil and the NO flux stays positive. In fact, NO deposition has been measured in other studies only in the case of high NO concentrations (>60ppb, Laville et al., 2011).

Figures 2 to 5 show daytime averaged NO and NH$_3$ fluxes ($\pm 1$ standard deviation) for each land cover type, along with precipitation and soil moisture. The spatial variability of NO fluxes is high, especially for bare soil, forest and the maize field where underground roots, not visible at the surface, are heterogeneously distributed. These roots are likely to influence the ammonium content of the soil, and the subsequent NO flux measurement. Standard deviation is generally smaller for grassland (except for two days, July 9th and 13th), where the vegetation (and the root distribution) is more homogeneous. The variation of soil moisture is consistent with the presence of rain events, showing a sharper increase of soil moisture at 5 cm, especially after rainfall following dry periods.

NO emissions from bare soil and grassland show an increase, sharper for grassland, one to two days after the rain event on 8[th] July. The longer rain event between 20[th] and 24[th] July does not seem to produce an increase in NO emissions (data available only for maize field and forest). This might be linked with the non-linear relationship between NO biogenic soil emissions and soil water content (Oswald et al., 2013). In fact, a light precipitation event (5-15 mm) occurring on dry soils can result in a large flux of NO (Meixner & Yang, 2006, Hartley & Sclesinger, 2000). However, when soil moisture stays at an equivalent level, after several rain events, pulse emissions do not occur (Millet et al., 2004). Due to this non linear character of the NO fluxes, no direct correlation was found between NO fluxes and environmental variables such as soil moisture or soil temperature taken individually. Moreover, soil temperature and soil moisture were not measured on the same soil parcel where the soil fluxes where measured and the location of the soil flux measurements was not kept constant even for the same land cover type on the same measurement day. This measurement protocol was designed to give an estimate of soil fluxes at a large ecosystem scale, rather than reproducing the relationships between soil fluxes and meteorological variables, like soil temperature and soil moisture.

A multiple linear regression analysis was performed between daytime mean NO fluxes and the following variables: wind speed, soil temperature at 5 cm, soil moisture at 5 cm, soil heat flux, upward longwave radiation and downward shortwave radiation. This regression gives $R^2$=0.49 (p-value=0.004), indicating a weak but existing relationship between those variables

and NO soil emissions, while the regression was weak between NO fluxes and each individual variable. This correlation

shows the influence of these environmental variables considered collectively on NO fluxes, highlighting the underlying mechanisms responsible for NO release to the atmosphere.Our experiment does not show the details of microbial and physical processes driving soil fluxes at a single point because measurements are done at different locations every day, but aims to estimate the spatial variability of fluxes at the ecosystem scale.

The NO flux estimated in this study does not consider the impact of vegetation on the net ecosystem flux, as we focus on soil fluxes only. However, the net emission to the atmosphere should take into account the oxidation of NO to NO2 and the eventual re-deposition of NO2 on the vegetation, i.e. what is called Canopy Reduction Factor and is assumed to be a linear function of the Leaf Area Index (e.g. Yienger and Levy, 1995, and Ganzeveld et al., 2002).

### 3.5 $NH_3$ fluxes

$NH_3$ fluxes measured during the field campaign range between -6.59 and 4.96 ngN $m^{-2}$ $s^{-1}$. Ammonia fluxes averaged over all land cover types are -0.91 $\pm$ 1.27 ngN $m^{-2}$ $s^{-1}$, showing a predominance of $NH_3$ deposition over emission, which is verified for every land cover type, with an average value of: -1.33 $\pm$ 0.86 ngN $m^{-2}$ $s^{-1}$ for bare soil, -0.75 $\pm$ 0.31 ngN $m^{-2}$ $s^{-1}$ for the maize field, -0.48 $\pm$ 0.55 ngN $m^{-2}$ $s^{-1}$ for grassland, and -0.30 $\pm$ 0.38 ngN $m^{-2}$ $s^{-1}$ for forest (Table 6). Low positive ammonia fluxes, indicating average $NH_3$ emission, are only recorded during three days, between $6^{th}$ and $8^{th}$ July, after the

longest dry period of the measurement campaign (Figures 2 to 5).
As discussed in Appendices B and C, of 30% of individual fluxes used to calculate the daily averages are very low and not distinguishable from adsorption or desorption of $NH_3$ on chamber walls. These very low fluxes are however meaningful and indicate that some periods of near zero fluxes must be taken into account to represent the processes of exchanges in these ecosystems.

To our knowledge, $NH_3$ soil fluxes from West African wet savanna are not available in the scientific literature. In Delon et al. (2017) $NH_3$ soil fluxes measured in Dahra (15°24' N 15°25' W), Senegal, on a dry savanna ecosystem, show low fluxes with a predominance of $NH_3$ emission: 1.3 $\pm$ 1.1 ngN $m^{-2}$ $s^{-1}$, -0.1 $\pm$ 1.1 ngN $m^{-2}$ $s^{-1}$ and 0.7 $\pm$ 0.5 ngN $m^{-2}$ $s^{-1}$ over three different measurements campaigns. However, Sutton et al. (2007) shows how pre-cut grassland is characterized by $NH_3$ deposition, as

in our study, in contrast to post-cut grassland, which is marked by $NH_3$ emission. It is interesting to notice that the literature provides up to about 700 ngN $m^{-2}$ $s^{-1}$ $NH_3$ emission for fertilized Zea Mays fields (Walker et al., 2013) while in our study site $NH_3$ deposition was recorded for the maize field, which is not treated with mineral fertilizer.

As for NO concentrations, $NH_3$ concentrations are reported in fig. 2 to 5. Daytime means of $NH_3$ concentration vary from

520 nearly 0 to 12.46 ppb for all sites, and the average concentration is 4.42 $\pm$ 3.23 ppb during the whole campaign. Average $NH_3$

concentration is 6.28 ± 3.90 ppb for bare soils, 3.28 ± 1.79 ppb for grassland, 4.36 ± 3.99 for the maize field, and 3.68 ± 2.13 ppb for forest (Table 6). The largest deposition fluxes are found on bare soils, where the largest concentrations are measured.

A multiple linear regression analysis was performed between daytime mean $NH_3$ fluxes and the following variables: wind speed, soil temperature at 5 cm, soil moisture at 5 cm, soil heat flux, outgoing longwave radiation and incoming shortwave radiation). This regression gives a weak but existing relationship, with $R^2$=0.37 (p-value=0.03). This correlation highlights the link between $NH_3$ fluxes and relevant environmental parameters. However, the same considerations explained in Sect. 3.4 for NO emissions are also valid for the correlation between $NH_3$ fluxes and meteorological variables.

According to the current parameterization of soil ammonia emission potential (Sect. 2.7), high values of pH and $[NH_4^+]$ in the liquid phase will determine high values of $\Gamma_g$ indicating that the soil has a high propensity to emit $NH_3$. However, despite the high values of $\Gamma_g$ recorded, our measurement site remains a net sink for $NH_3$. The reasons for this can be manifold. One explanation could be that soil particles on our site may have a high adsorption capacity limiting the amount of soil gaseous $NH_3$ concentrations (Neftel et al., 1998) and the largest part of the estimated ammonium content in the soil may not be in the liquid phase, but adsorbed by solid soil particles. In these conditions ammonium will not be available for gas exchange to open porosity and the atmosphere (Flechard et al., 2013). Another explanation could be given by the presence of a water film at the soil surface (linked to high air humidity at the site), which will increase the net deposition process. David et al. (2009) conclude from their measurements that the bare soil can be a significant source of $NH_3$ only for a limited period and only when the cut vegetation is removed, but not if the soil surface remains covered by grass. Measurements in Ferrara et al. (2014) show other occurrences of high soil ammonia emission potential and $NH_3$ deposition.

Our measurements were conducted without vegetation inside the chambers, but vegetation was present in the fields. It is important to mention that the role of vegetation on $NH_3$ bidirectional fluxes is essential, especially during the wet season (time of the experiment), when deposition on the vegetation through stomata and cuticles dominate the exchange (during rain events, the cuticular resistance becomes small and cuticular deposition dominates), due to an increase of the deposition velocity of $NH_3$ (consecutive to the humidity response of the surface) and a decrease of the canopy compensation point, sensitive to the surface temperature and the surface wetness (Wichink-Kruit et al., 2007).

**3.6 Comparison of observed and modelled NO soil emissions**

We have compared observed daytime averaged (8 a.m. to 6 p.m.) soil NO emissions with those modelled by GEOS-Chem for the entire period of the campaign over the model grid box including the measurement site. The model grid box is positioned at latitude 8.0° N – 8.25° N and longitude 2.19° E – 2.5° E. The area of this grid box is 958 $km^2$. The land cover type within this grid box is classified as "Savannah (Warm)" but the surrounding area also consists of "Woody Savannah",

while the observations were taken over the four land cover types representative of the region: bare soil, grassland, maize field and forest.

The model is able to reproduce mean air temperature (25.3 ± 0.7 °C) and the main rain events. Soil emissions of NO are well simulated in magnitude. Simulated NO emissions are often higher than those recorded over the grassland areas, however, simulated NO emissions are often within the error bars of measurements (Fig. 6). The model uses land cover and vegetation types to simulate the highly variable land and vegetation cover of the observation site, for this reason we do not expect the

560 model to reproduce the site-to-site variability of the measured soil fluxes, but to at least reproduce their average magnitude and behaviour..It appears that when the model is able to reproduce the length and the intensity of the rain events, NO emissions are especially well simulated, e.g. the model is able to reproduce the longest rain period (from 20[th] to 26[th] July 2016) and the decrease of emissions at the end of the measurement campaign.

### 3.7 Estimate of total NO soil emissions and $NH_3$ deposition for Benin

In order to give a tentative estimate of NO and $NH_3$ soil fluxes for Benin we have used the land use/land cover map of Benin provided by the US Geographical Survey Atlas: Landscapes of West Africa – A Window on a Changing World (CILSS, 2016, Fig. 7a). The method of mapping land use/land cover used in this atlas was based on Landsat imagery and expert visual interpretation. In particular, these maps provide an accurate indication of cropland distribution using visual interpretation. According to CILSS (2016) Benin's present-day (2013) land cover is mainly savanna, almost 60%, followed by agricultural

land, 31%, while forest is only a small fraction under 1% (the rest of the surface is mainly gallery forest and, on a smaller extent, settlements). In the Atlas (CILSS, 2016) bare soils are defined as those surfaces that are bare even in the green/rainy season. For Benin, the amount of bare soil estimated by CILSS (2016) is very small, not big enough to appear on the Atlas' maps. We have multiplied average NO emissions measured at the Savè site for each land cover type by an estimate of the land cover area of each class given by the Atlas. We have made some approximations: as the land use/land cover maps do not

distinguish between shrub savanna, tree savanna, and wooded savanna, we have considered NO soil emissions from Savè's grassland savanna to be representative of the general savanna category in CILSS (2016). Moreover, the Atlas has a crop category that does not distinguish the type of crop and we only have observations of NO soil emissions from an intercropped maize field. We have taken NO soil emissions from the maize field as representative of NO emissions of Benin's agricultural land, but other cultures are present in other parts of the country, e.g. oil palm plantations, where possibly stronger

fertilization could determine higher NO soil emissions. This tentative calculation gives that Benin's NO soil emissions for the month of July (wet season) is 1.17 ± 0.6 GgN/month, i.e. 0.09% of the average global monthly NO soil emissions as given by Davidson et al., (1997).

We have also calculated Benin's total monthly NO soil emissions with GEOS-Chem adding together the NO soil emissions

from the grid boxes where 50% or greater of the box lies within Benin. Benin's total monthly NO soil emissions calculated

with GEOS-Chem for the month of July are 1.44 GgN/month and agree with the tentative calculation given above (1.54 ± 0.8 GgN/month). However, the land cover types covering Benin in GEOS-Chem differ from those in the US Geographical Survey Atlas (CILSS, 2016). In GEOS-Chem Benin is cover by 60.9% savanna, 31.4% woody savannah, 4.5% grassland, 1.3% mixed forest and 0.6% urban and built-up lands. Benin's total monthly NO soil emissions calculated with GEOS-Chem for the months of May and June are higher, 3.51 GgN/month and 2.59 GgN/month, respectively, given those months are at the beginning of the wet season and are characterized by more predominant pulse emissions. Using the same method described above we have upscaled point measurements of $NH_3$ fluxes with relevant land cover surfaces from CILSS (2016) and obtained that total $NH_3$ dry deposition for the month of July is 0.21 ± 0.11 GgN/month (0.22 kgN ha$^{-1}$ yr$^{-1}$). This value is about ten times smaller than the estimation of $NH_3$ dry deposition given in Adon et al. (2013) for the wet savanna site of Djougou (Benin, 9.7°N, 1.7°E) for the month of July, which is around 2.5 kgN ha$^{-1}$ yr$^{-1}$.

## 4. Conclusion

We provide soil flux measurements along with soil characteristics for a land cover type, savanna, that is considered to have large NO emissions (Davidson and Kingerlee, 1997), and for an area of the world, West Africa, with little observations. The aim of this study is to contribute to our knowledge in biogenic soil nitrogen exchanges, provide data for inventories and model evaluation to improve air quality and climate modelling.

In situ measurements were made in a wet savanna site in central Benin from mid-June to the end of July 2016. Complementary to these exchange fluxes, soil N and C content, as well as soil pH, soil moisture, soil temperature and meteorological data were measured. Soil fluxes of NO and $NH_3$ were measured over four different land cover types in order to give a tentative estimate of regional soil fluxes.

Given the set up of the experiment, the known relationships between soil fluxes, soil temperature and soil moisture were not reproduced. Rather than looking at the microbial and physical processes behind soil fluxes, we are able to provide observations that are representative of a bigger surface area and that represent the spatial variability of fluxes. However, we observe that while shorter rain events determine an increase in NO soil emissions, the longer rain event at the end of the campaign (20[th] to 24[th] July 2016) is accompanied by a decrease in NO soil emissions, in agreement with the fact that the relationship between NO soil emissions and soil moisture is not univocal. Soil emissions of NO increase until an optimum value of soil moisture is reached and then decrease (Oswald et al., 2013).

$NH_3$ emissions measured in this study probably underestimate total $NH_3$ emissions for the entire country, as possibly higher localized $NH_3$ emissions are present in the south of the country where industrial scale agriculture would probably deploy mineral N fertilization.

Soil NO emissions simulated by GEOS-Chem are in good agreement with the local observations taken at the site of Savè, providing a good baseline for simulating local atmospheric chemistry. Moreover, GEOS-Chem is also in good agreement with the tentative total monthly NO soil emission estimate for Benin for the month of July made with local observation in Savè and US Geographical Survey Atlas (CILSS, 2016). All these elements contribute to improve our confidence in the results of modelling studies of local and regional air quality and climate over this region.

Agriculture is the first form of economic activity in Benin, occupying a majority of the active population. The most obvious recent change in land cover is the major expansion of agricultural land across most regions of Benin. Agricultural areas (including plantations and irrigated agriculture) progressed from 9.2 to 27.1 % of the total country area between 1975 and 2013, improving food security. Oil palm trees are the main crop, and oil palms farmland already covered most of the southern Terre de Barre plateau of Benin by 1975, and increased by about 28 percent over the following 38-year period. A century or more ago, Benin was covered by dense, biologically diverse forest. Since then, Benin has lost nearly all of that forest cover, by 2013, 58 % of the 1975 forest cover had been lost, leaving only 0.2 percent of the country covered with dense forest. Savanna area has also decreased by 23 percent since 1975, but it still remains the dominant land cover type in Benin and covers more than half of the country (CILSS, 2016).

More measurements of NO and $NH_3$ exchanges between soil-vegetation-atmosphere in areas of Benin (or West Africa) interested by land-use change could improve our estimate of the impact of biogenic soil emissions on air quality and climate, as biogenic soil fluxes influence for example the amount of aerosol and tropospheric $O_3$, a greenhouse gas and pollutant, in the atmosphere. Management practices of agriculture affect biogenic soil emissions. Moreover, loosing savanna to oil palm plantations or other crop would have different impacts on air quality, carbon budget and climate than the conversion of forest into crop or oil palm plantation. Furthermore, oil palm plantations are generally closer to the coast and likely to be more influenced by anthropogenic emissions from industry and coastal cities (Knippertz et al., 2015a, 2015b). Oil palm trees are also a strong isoprene emitters. Isoprene emissions influence ozone concentration and the oxidizing capacity of the atmosphere, and it is a source of secondary organic aerosol, thus affecting local air quality and global climate. Large-scale land use change in the tropics – specifically the conversion of tropical rain forest to oil palm plantations in Malaysia – were shown to cause changes in atmospheric composition and chemistry (Hewitt et al., 2009), indicating that the management of the emissions of reactive nitrogen species is essential to prevent damaging levels of ground-level ozone in those regions.

 Appendix A

The dilution uncertainty is calculated based on the uncertainties of standard concentration, standard flow and dilution flow. The uncertainty of standard concentration is 5% for NO and $NO_2$, 2% for $NH_3$. The maximum uncertainty of dilution flow is 1% of the plain scale (10 L.min$^{-1}$) for the three standards divided by the flow used in the diluter (3.2 L.min$^{-1}$ maximum), which gives 0.1/3.2=3.1%. The uncertainty of standard flow is 1% of the plain scale standard flow (50 mL.min$^{-1}$) divided by the standard flow used to obtain the needed concentration (50 ppb for NO or 30 ppb for $NH_3$).

Standard flow=(needed concentration/standard concentration)*dilution flow.

**For NO**, dilution flow=3.2 L.min$^{-1}$, needed concentration=50 ppb, standard concentration=8.73 ppm, standard flow=18.4 mL.min$^{-1}$. Uncertainty of the standard flow = 1%*50/18.4=2.71%. Total uncertainty is therefore 5%+3.1%+2.7%=**10.8%.**

**For $NO_2$**, dilution flow=3.2 L.min$^{-1}$, needed concentration=50 ppb, standard concentration=9.28 ppm, standard flow=17.2 mL.min$^{-1}$. Uncertainty of the standard flow = 1%*50/17.2=2.9%. Total uncertainty is therefore 5%+3.1%+2.9=**11%**

**For NH3**, dilution flow=3.2 L.min$^{-1}$, needed concentration=30 ppb, standard concentration=14.78 ppm, standard flow=6.5 mL.min$^{-1}$. Uncertainty of the standard flow =1%*50/6.5=7.7%. Total uncertainty is therefore 2%+3.1%+7.7%=**12.8%.**

Appendix B

We ran a laboratory experiment to verify that deposition on the walls of the Teflon chamber is negligible.

Ambient air concentrations were measured by the analyzer, inside the room where the analyzer and the chamber were placed. Measurements of $NH_3$ concentrations were made in ambient air with and without the Teflon chamber attached to the analyzer. The Teflon chamber was placed on a Teflon frame, and they were sealed together with Teflon tape. Measurements of $NH_3$ concentrations with the Teflon chamber attached to the analyzer were followed by measurements without the chamber 30 to 60 minutes later. The two sets of measurements were made under similar conditions of temperature and humidity. Average values of $NH_3$ concentrations were calculated for 10 to 30 minutes before and after connecting the chamber. Average $NH_3$ concentrations during this time interval varied between 8 and 36 ppb, with a variation between 1.5 to 13% around the mean. The lowest $NH_3$ concentrations correspond to air samples previously passed through charcoal and desiccant cartridges (NO and $NO_2$ zero air). Measured $NH_3$ concentrations are reported in Table B1, along with temperature, humidity and the ratio between average concentration with and without the Teflon chamber attached to the analyzer.

This test was made at different times of the day on different days: air humidity varied between 46 and 54%, temperature varied between 25 to 29°C, while pressure varied between 1006 and 1008 hPa (not reported).

Results show negligible variation between concentrations of air reaching the analyzer via the chamber or going directly to the analyzer; The average difference in concentration is 0.9 ppb, and should be considered as the detection limit for fluxes

significantly different from zero (i.e. including potential effects of adsorption or desorption). 30% of the concentration differences are below 0.9 ppbv.

| Temperature (°C) | Humidity (%) | [NH₃] TC | [NH₃] D | Ratio [NH₃]TC / [NH₃]D |
|---|---|---|---|---|
| 25 | 54 | 26.4±0.4 | 25.9±0.5 | 1.02 |
| 27 | 49 | 9.0±0.8 | 8.0±1.2 | 1.12 |
| 29 | 46 | 35.2±2.4 | 36.0±1.4 | 0.98 |
| 28 | 46 | 25.6±0.7 | 24.0±1.0 | 1.07 |
| 27 | 45 | 26.2±0.9 | 27.4±0.9 | 0.96 |
| 28 | 46 | 23.4±0.3 | 24.1±0.4 | 0.97 |
| 26 | 50 | 19.2±0.6 | 18.4±0.8 | 1.04 |

Table B1   Measurements of $NH_3$ concentrations (ppb) through the chamber (TC) or directly (D) to the analyzer.

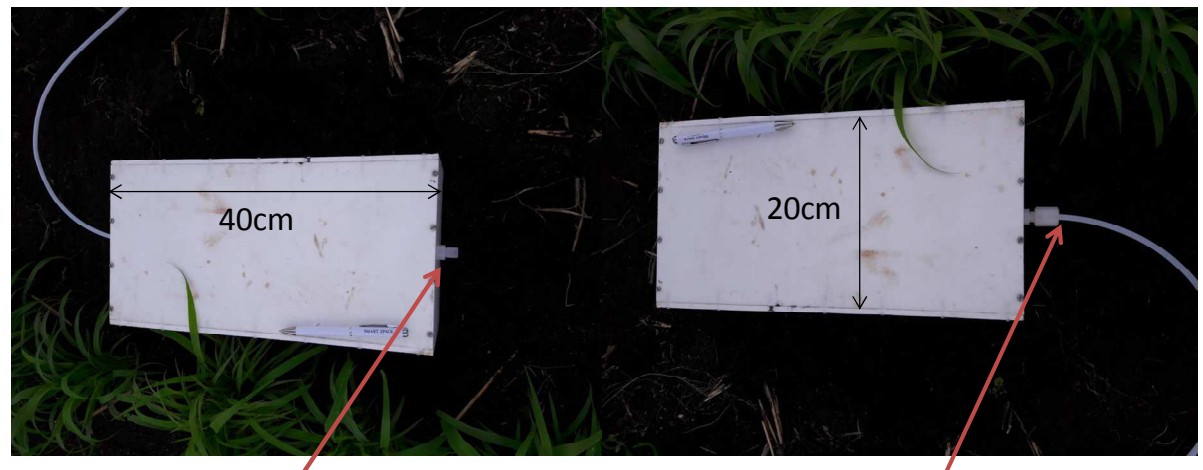

Vent (4mm diameter)          Teflon tube connected to the analyzer

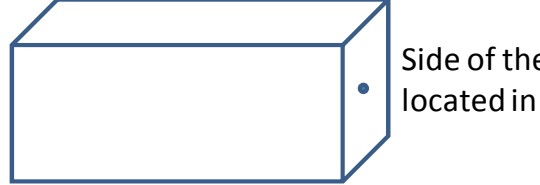

Side of the chamber: vent and tube connection located in the middle of the square

Picture B1: Description of the Teflon chamber.

To verify if mixing the air in the chamber by a fan would change the shape of the increase in concentration or the concentration itself in the chamber, a test was made with a syringe simulating the action of a fan (i.e. we have mixed the air

inside the chamber by sucking and releasing the same air with a syringe through the small vent, while letting outside air entering the chamber by the vent as usual to ensure pressure equilibrium between outside and inside air). The comparison between a flux measurement with and without mixing gives similar slopes of the concentration increase (or decrease).

Appendix C

$NH_3$ concentration was recorded continuously inside the Teflon chamber (placed on a Teflon material as in the tests summarized in Appendix A), and the chamber was cleaned successively with three different papers, referred to as A, B and C. The concentration was recorded at least for 30 minutes between every cleaning. Table C1 summarizes the averaged concentrations (and standard deviations) for every period. Results show a variation of concentration when different papers are used, but this variation is not reproducible and is difficult to differentiate from a natural variation of the $NH_3$ concentration in the room. As a matter of fact, the effect of cleaning on $NH_3$ adsorption or desorption is not clear, but questions about the potential pollution of the chamber arise. Results in Table C1 may lead to the conclusion that if the difference in concentration during a flux measurement is inferior to a certain threshold, it is not necessarily a flux from the ground but could be due to adsorption or desorption of $NH_3$ by the chamber walls due to cleaning. Only low fluxes are concerned. Positive or negative fluxes inferior to 0.55 $ngN.m^{-2}.s^{-1}$ (corresponding to concentration differences less than 0.9 ppb in the flux calculation, as defined in Appendix B) represent 30% of the 350 measured fluxes. If those low fluxes were removed from the database, the resulting average would be slightly larger in magnitude (-1.1 instead of -0.9 $ngN.m^{-2}.s^{-1}$). As a conclusion, these tests may help to warn the reader that caution must be kept for low $NH_3$ fluxes because of possible pollution in the chamber.

| Paper used for cleaning | 30 minutes average (standard deviation) in ppb | Difference between two successive averages in ppb |
|---|---|---|
| First day of test | | |
| Before cleaning | 8.13±0.58 | |
| A | 7.43±0.47 | -0.69 |
| B | 8.55±0.79 | 1.12 |
| C | 9.73±0.81 | 1.18 |

| B | 9.75±0.88 | 0.02 |
|---|---|---|
| Second day of test | | |
| B | 16.35±0.92 | |
| C | 16.77±0.60 | 0.42 |
| A | 17.30±0.76 | 0.53 |
| A | 18.94±0.72 | 1.64 |
| B | 18.96±0.62 | 0.02 |
| C | 18.79±0.79 | -0.17 |

Table C1: 30 minutes averaged concentrations in the Teflon chamber after cleaning with different dry papers.

Appendix D

| Location of the temperature measurement | Temperature (°C) |
|---|---|
| Air | 32.7 |
| Soil | 34.2 |
| Chamber: outside wall | 33.1 |
| Chamber: inside wall | 33.3 |
| Chamber: outside top | 33.8 |
| Tube: outside close to the chamber | 30.5 |
| Tube : outside close to the analyzer | 30.3 |
| Tube: inside | 30.9 |

Table D1 :  Temperature measured on the Teflon chamber and on the Teflon tube. These measurements have been made after the field campaign in direct sunlight at 3:30 PM. Measurements were made with a calibrated thermometer  HI 98509 with stainless steel probe (-50 → +150 °C).

Appendix E

NO and $NH_3$ fluxes are calculated from the slope of their concentration increase (or decrease) in the chamber through time.
Two examples are given in figures E1 to illustrate the larger instability of $NH_3$ detection compared to NO detection, due to
possible interaction of $NH_3$ with chamber walls, or particulate matter in the chamber.

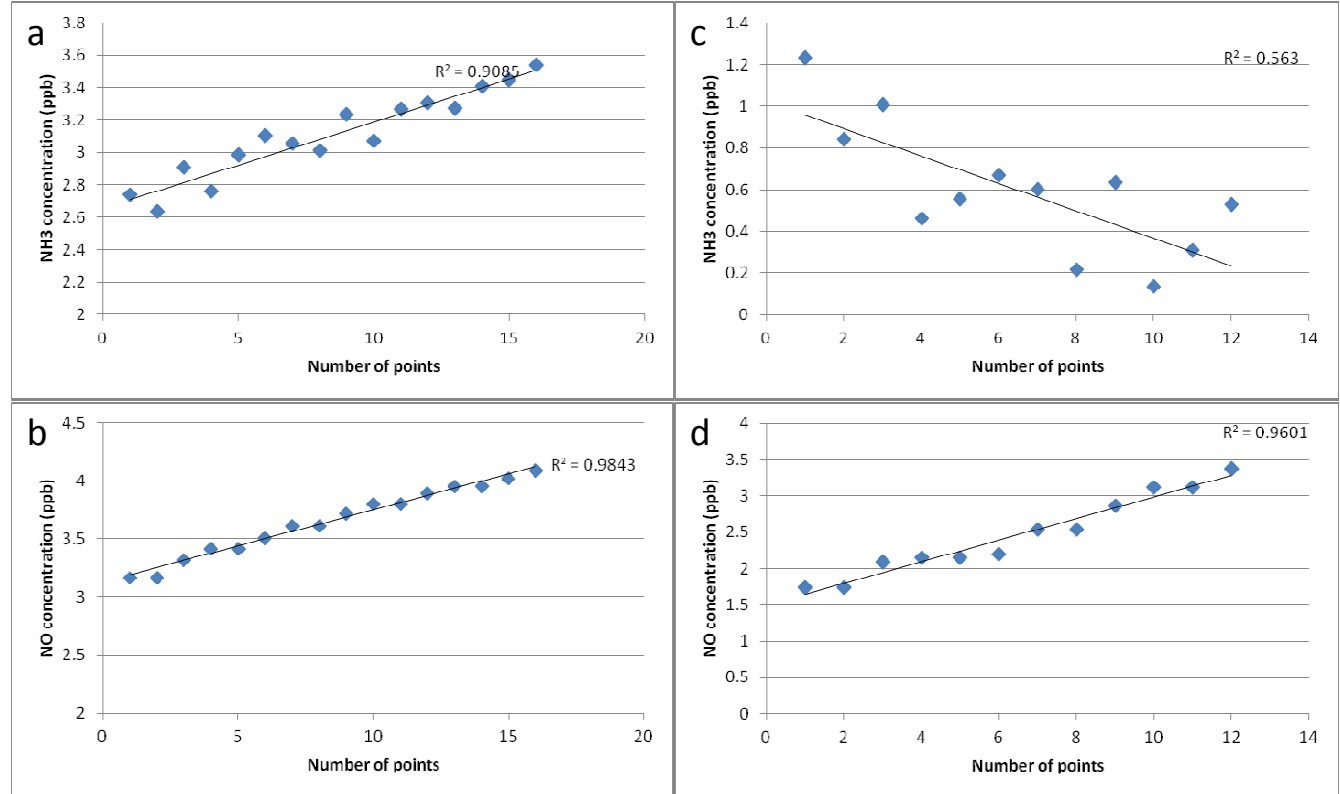

Figure E1: NH$_3$ (a,c) and NO concentration (b,d) variation with time (one point every 10 second) inside the chamber on grassland (left) and bare soil (right).

**Acknowledgements**

The DACCIWA project has received funding from the European Union Seventh Framework Programme (FP7/2007-2013) under grant agreement no. 603502. We also want thank the staffs from NCAS (National Centre for Atmospheric Science), KIT (Karlsruhe Institute of Technology) and UPS (Université Paul Sabatier, Toulouse III) for helping to install the equipment as well as the people from INRAB in Savè for allowing the equipment on their ground.

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

| | Savè ground-based observation site |
|---|---|
| Location | 8°02'03" N, 2°29'11" E |
| Elevation | 166 m a.s.l. |
| Mean annual precipitation | 1100 mm |
| Mean annual temperature | 27.5 °C |
| Soil type | sandy |
| Sand percentage | 87% |
| Clay percentage | 4.1% |

Table 1

Main characteristics of the Savè site.

| Soil type | | Plant family | Plant species | Common name/s |
|---|---|---|---|---|
| Next to grassland and forest | Dominant tree species | Anacardiaceae | *Anacardium occidentale* | cashew tree |
| | | Fabaceae | *Daniellia oliveri* | African copaiba balsam tree |
| | | | *Pterocarpus erinaceus* | barwood, muninga, vène, mukwa |
| | Dominant ground species | Cleomaceae | *Cleome* sp. | spider flowers, spider plants |
| | | Fabaceae | *Crotalaria* sp. | rattlepod or rattlebox |
| | | | *Mucuna* sp. | velvet bean |
| | | Poaceae | *Imperata cylindrica* | cogon grass, cotton wool grass, kura-kura |
| | | | *Rhynchelytrum repens* | rose natal grass |
| Next to maize field | Dominant tree species | Anacardiaceae | *Mangifera indica* | mango |
| | | Arecaceae | *Cocos nucifera* | coconut tree |
| | | Caricaceae | *Carica papaya L.* | papaya |
| | | Lamiaceae | *Tectona grandis* | teak |
| | | Meliaceae | *Azadirachta indica* | neem, nimtree, Indian lilac |
| | Dominant ground species | Commelinaceae | *Commelina benghalensis* | benghal dayflower, tropical spiderwort |
| | | Euphorbiaceae | *Euphorbia* sp. | spurge |
| | | Nyctaginaceae | *Boerhavia diffusa* | punarnava, red spiderling |
| | | Phyllanthaceae | *Phyllanthus amarus* | gale of the wind, stonebreaker |
| | | Poaceae | *Digitaria horizontalis* | Jamaican crabgrass |
| | Crops | Dioscoreaceae | *Dioscorea* sp. | yam |
| | | Euphobiaceae | *Manihot esculenta* | cassava |
| | | Fabaceae | *Arachis hypogaea* | peanut |
| | | | *Vigna unguiculata* | cowpea |
| | | Malvaceae | *Gossypium* sp. | cooton |
| | | Pedaliaceae | *Sesamum indicum* | sesame |
| | | Poaceae | *Zea mays* | maize |
| | | | *Sorghum* sp. | sorghum |
| | | Solanaceae | *Solanum lycopersicum* | tomato |

Table 2

List of plant species at the Savè site. The list of common names is not considered to be exhaustive.

| | Bare Soil | Grassland | Maize Field | Forest |
|---|---|---|---|---|
| Clay (<2μm) (%) | 5.13±0.63 | 3.15±0.50 | 4.40±0.35 | 3.70±1.25 |
| Fine Silt (2 to 20μm) (%) | 5.13±0.96 | 2.93±0.32 | 4.13±1.00 | 3.40±1.21 |
| Coarse Silt (20 to 50μm) (%) | 5.98±0.51 | 4.78±0.66 | 4.37±0.38 | 4.67±1.05 |
| Total Sand (50 to 2000 μm) (%) | 83.75±1.82 | 89.20±0.71 | 87.13±0.99 | 88.20±3.50 |

Table 3

List of soil characteristics for each land cover type at the Savè site, including standard deviation.

| Soil Type | Date | C/N ratio | Organic C (g kg$^{-1}$) | Total N (g kg$^{-1}$) |
|---|---|---|---|---|
| Bare soil | 06/07/2016 | 14.90 | 24.28 | 1.63 |
| | 09/07/2016 | 12.20 | 10.47 | 0.86 |
| | 19/07/2016 | 12.30 | 15.05 | 1.22 |
| | 28/07/2016 | 9.50 | 19.30 | 2.04 |
| Grassland | 07/07/2016 | 15.20 | 5.78 | 0.38 |
| | 09/07/2016 | 16.00 | 7.36 | 0.46 |
| | 19/07/2016 | 14.90 | 7.16 | 0.48 |
| | 28/07/2016 | 10.90 | 4.56 | 0.42 |
| Maize | 09/07/2016 | 14.80 | 17.33 | 1.17 |
| | 19/07/2016 | 16.40 | 13.08 | 0.8 |
| | 28/07/2016 | 11.80 | 11.83 | 1.00 |
| Forest | 06/07/2016 | 14.80 | 7.98 | 0.54 |
| | 19/07/2016 | 11.90 | 9.62 | 0.81 |
| | 28/07/2016 | 11.30 | 16.56 | 1.47 |

Table 4

List of soil characteristics for each land cover type at the Savè site for each soil sampling day: Carbon-to-Nitrogen ratio (C/N), organic carbon (g kg$^{-1}$) and total nitrogen (g kg$^{-1}$). The accuracy for the C/N ratio is 14%. The measurement accuracy for organic carbon and total nitrogen is 14 and 13%, respectively.

| Soil Type | Date | pH | $[NH_4^+]$ (mg kg$^{-1}$) | $\Gamma_g[NH_4^+]/[H^+]$ | $X_g$ (ppb) |
|---|---|---|---|---|---|
| Bare soil | 06/07/2016 | 8.38 | 6.82 | 136 334 | 1891 |
|  | 09/07/2016 | 7.73 | 2.90 | 12 978 | 134 |
|  | 19/07/2016 | 8.46 | 6.63 | 159 343 | 2188 |
|  | 28/07/2016 | 8.34 | 8.01 | 146 033 | 2215 |
| Grassland | 07/07/2016 | 7.15 | 1.96 | 2 307 | 29 |
|  | 09/07/2016 | 7.91 | 1.55 | 10 499 | 108 |
|  | 19/07/2016 | 7.52 | 2.28 | 6 291 | 86 |
|  | 28/07/2016 | 6.51 | 2.30 | 620 | 9 |
| Maize | 09/07/2016 | 7.46 | 4.40 | 10 575 | 109 |
|  | 19/07/2016 | 7.61 | 14.74 | 50 040 | 687 |
|  | 28/07/2016 | 8.04 | 4.49 | 41 027 | 622 |
| Forest | 06/07/2016 | 6.32 | 2.18 | 380 | 5 |
|  | 19/07/2016 | 7.51 | 4.88 | 13 159 | 181 |
|  | 28/07/2016 | 7.37 | 11.47 | 22 407 | 340 |

Table 5

List of soil pH, ammonium concentrations $[NH_4^+]$ (mg kg$^{-1}$), soil emission potential $\Gamma_g$ and soil compensation point $\chi_g$ (ppb) for each land cover type at the Savè site for each soil sampling day. The measurement accuracy for pH is 0.15 when pH $\leq$ 7 and 0.20 when pH > 7. The accuracy for ammonium concentrations $[NH_4^+]$, soil emission potential $\Gamma_g$ and soil compensation point $\chi_g$ is 25%.

| | Bare soil | Grassland | Maize field | Forest | Mean over all land cover types |
|---|---|---|---|---|---|
| NO fluxes (ng.m$^{-2}$.s$^{-1}$) | 8.05 ± 3.49 | 2.82 ± 3.46 | 3.73 ± 1.76 | 2.87 ± 1.49 | 4.79 ± 5.59 |
| NH$_3$ fluxes (ng.m$^{-2}$.s$^{-1}$) | -1.33 ± 0.86 | -0.48 ± 0.55 | -0.75 ± 0.31 | -0.30 ± 0.38 | -0.91 ± 1.27 |
| NO concentration (ppb) | 2.97 ± 1.49 | 2.57 ± 0.96 | 2.55 ± 0.83 | 2.76 ± 0.65 | 2.70 ± 1.03 |
| NH$_3$ concentration (ppb) | 6.28 ± 3.90 | 3.28 ± 1.79 | 4.36 ± 3.99 | 3.68 ± 2.13 | 4.42 ± 3.23 |

Table 6: List of average NO and NH$_3$ fluxes (ngN.m$^{-2}$.s$^{-1}$) and concentrations (ppb) for bare soil, grassland, maize field and forest sites, and for all cover types.

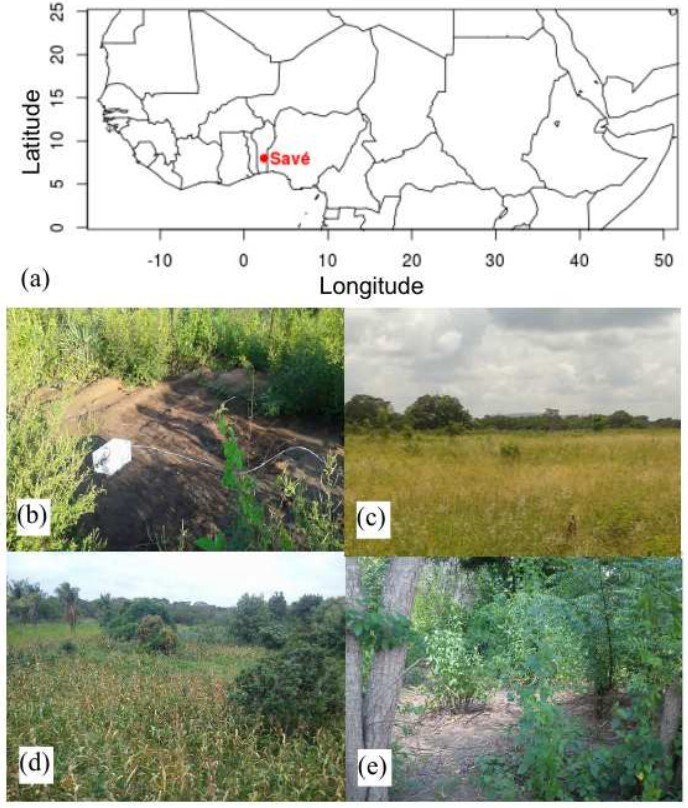

Fig. 1(a) Location of the Savè site in West Africa, (b) one of the bare soil sampling sites, (c) the grassland sampling site, (d) the maize field sampling site and (e) the forest sampling site at the Savè site.

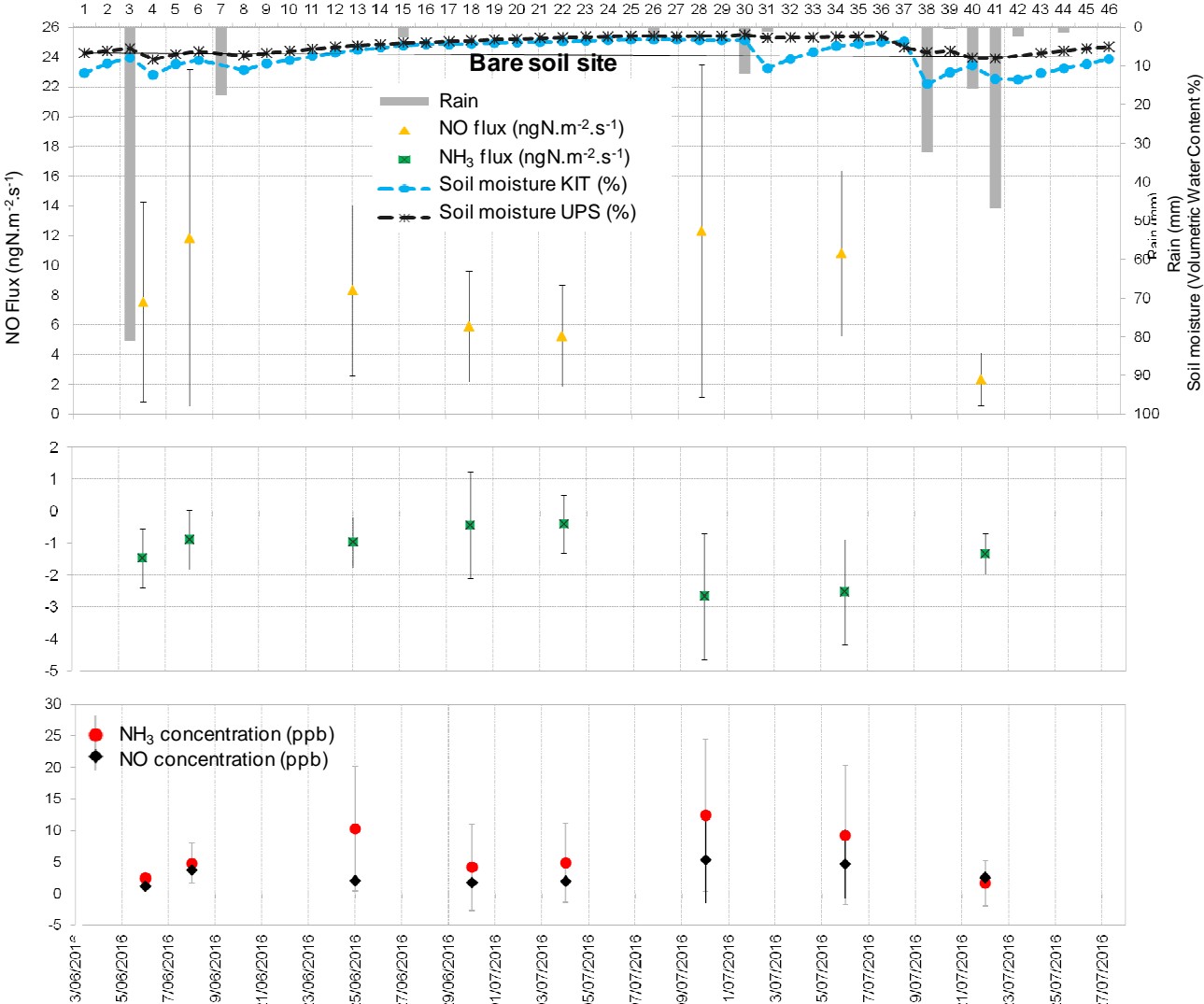

Fig. 2 Upper panel: Daily total precipitation (mm), daily mean soil moisture at 5 cm (%) measured by the Karlsruhe Institute of Technology (KIT), daily mean soil moisture averaged between 0 and 30 cm measured by the Université Paul Sabatier (UPS) instrumentation, daytime mean NO fluxes in ngN m$^{-2}$ s$^{-1}$ measured at the bare soil site; Middle panel: daytime mean NH$_3$ fluxes in ngN m$^{-2}$ s$^{-1}$ ; Lower panel:  daytime mean NO and NH$_3$ concentrations in ppb. Vertical bars show the standard deviation from individual fluxes and concentrations

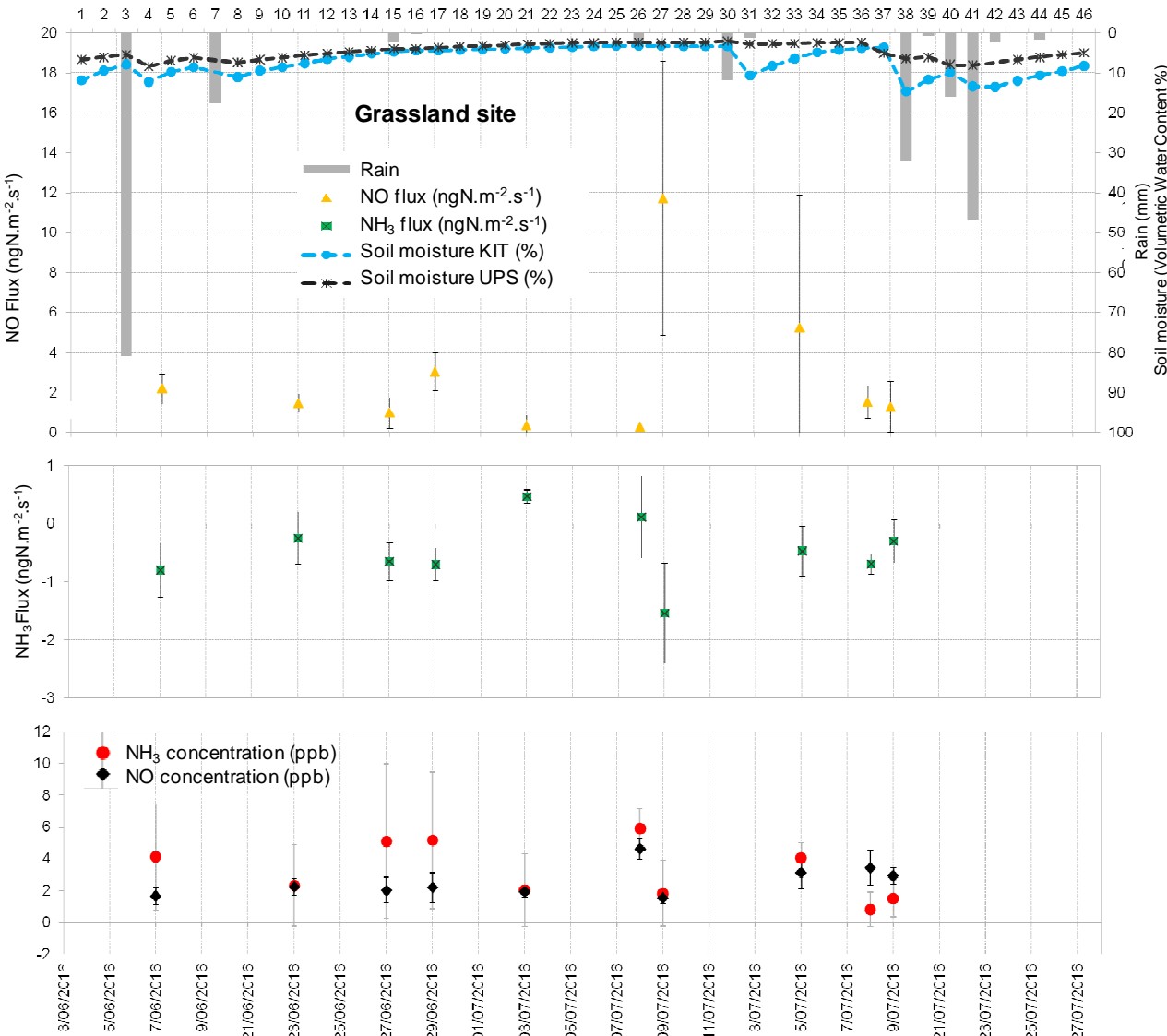

Fig. 3 Upper panel: Daily total precipitation (mm), daily mean soil moisture at 5 cm (%) measured by the Karlsruhe Institute of Technology (KIT), daily mean soil moisture averaged between 0 and 30 cm measured by the Université Paul Sabatier (UPS) instrumentation, daytime mean NO fluxes in ngN m$^{-2}$ s$^{-1}$ measured at the grassland site; Middle panel: daytime mean NH$_3$ fluxes in ngN m$^{-2}$ s$^{-1}$; Lower panel: daytime mean NO and NH$_3$ concentrations in ppb. Vertical bars show the standard deviation from individual fluxes and concentrations.

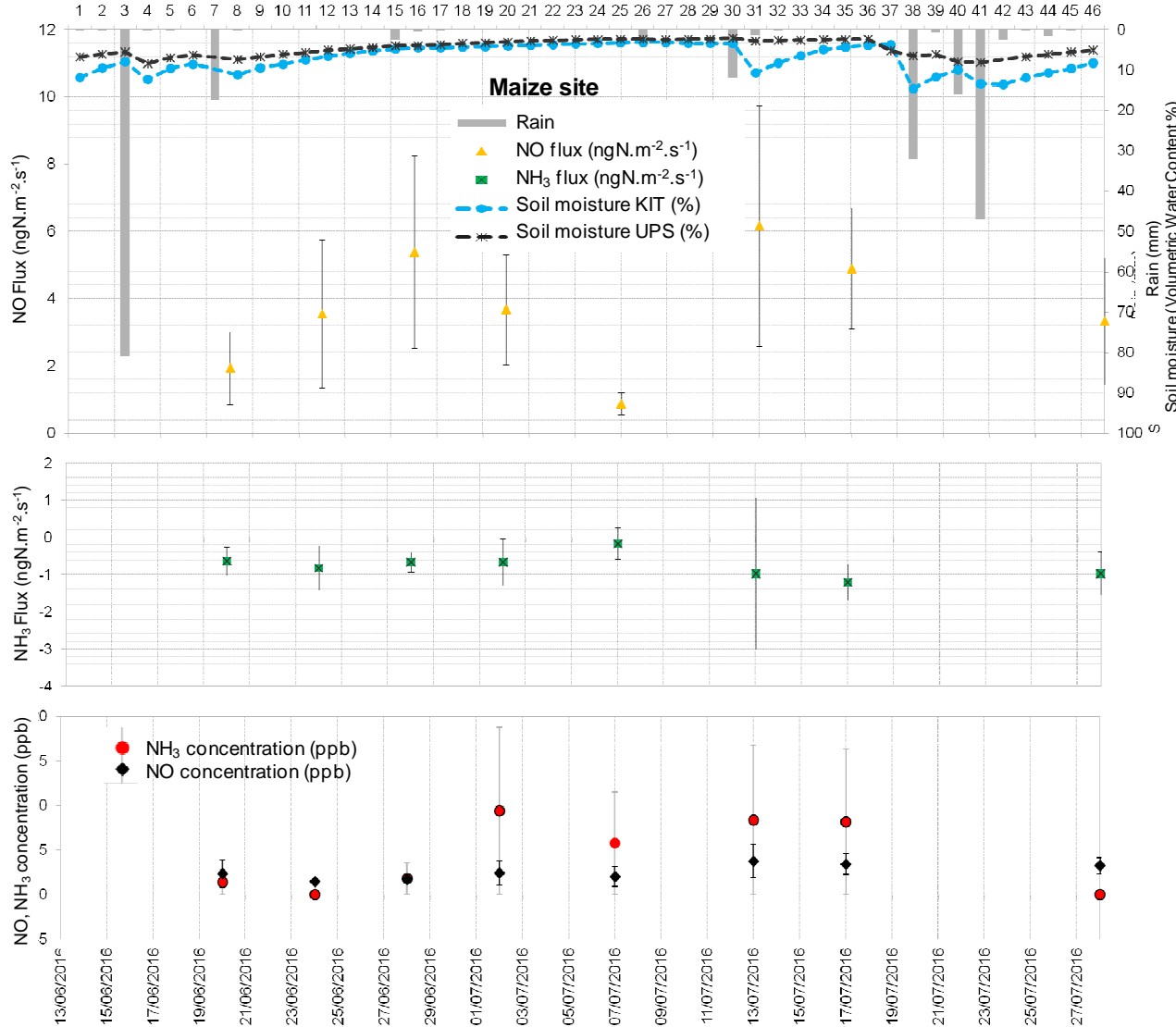

Fig. 4 Upper panel: Daily total precipitation (mm), daily mean soil moisture at 5 cm (%) measured by the Karlsruhe Institute of Technology (KIT), daily mean soil moisture averaged between 0 and 30 cm measured by the Université Paul Sabatier (UPS) instrumentation, daytime mean NO fluxes in ngN m$^{-2}$ s$^{-1}$ measured at the maize field site ; Middle panel: daytime mean NH$_3$ fluxes in ngN m$^{-2}$ s$^{-1}$; Lower panel: daytime mean NO and NH$_3$ concentrations in ppb. Vertical bars show the

standard deviation from individual fluxes and concentrations.

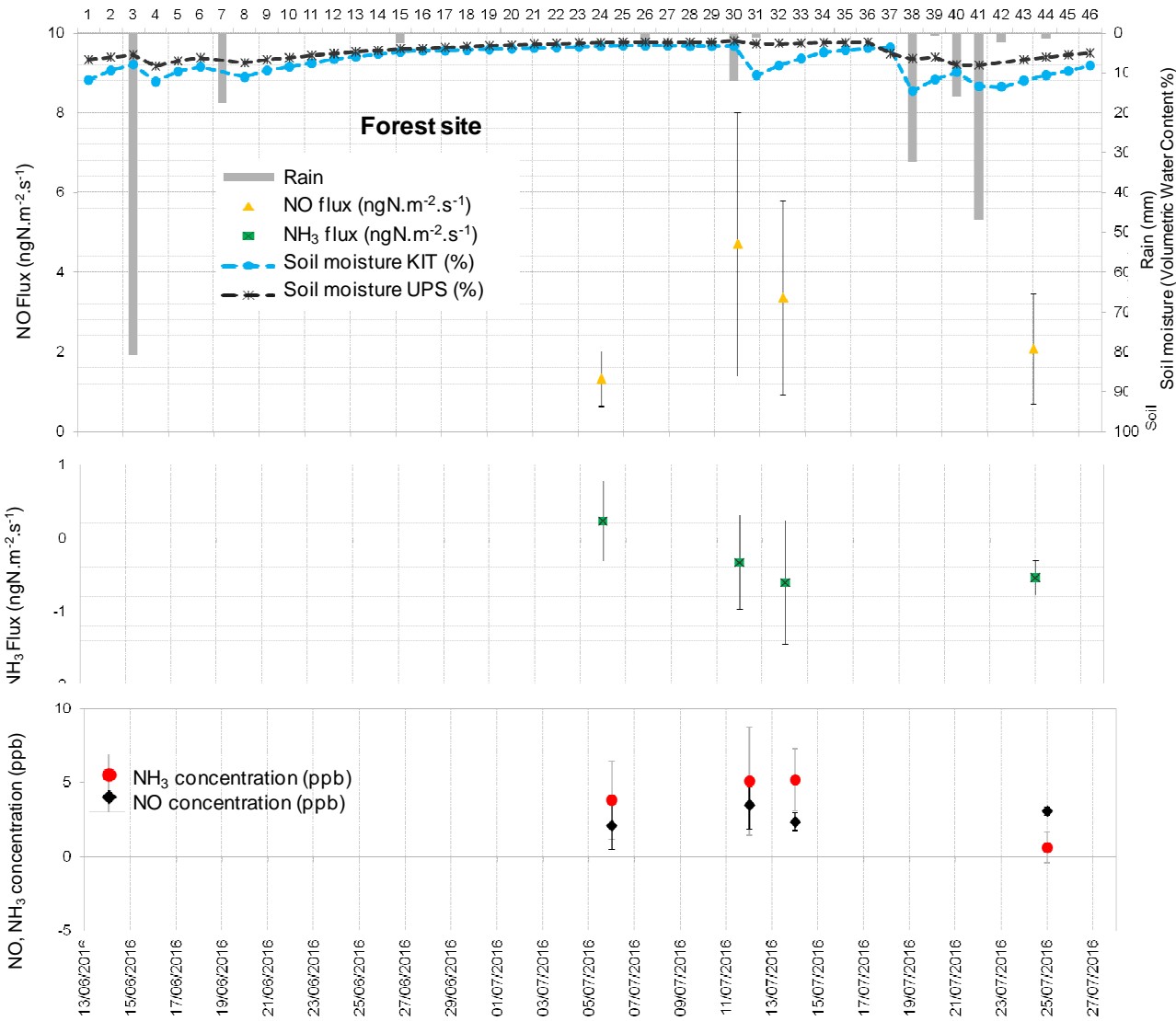

Fig. 5 Upper panel: Daily total precipitation (mm), daily mean soil moisture at 5 cm (%) measured by the Karlsruhe Institute of Technology (KIT), daily mean soil moisture averaged between 0 and 30 cm measured by the Université Paul Sabatier (UPS) instrumentation, daytime mean NO fluxes in ngN $m^{-2}$ $s^{-1}$ measured at the forest site ; Middle panel: daytime mean NH$_3$ fluxes in ngN $m^{-2}$ $s^{-1}$; Lower panel: daytime mean NO and NH$_3$ concentrations in ppb. Vertical bars show the standard deviation from individual fluxes and concentrations.

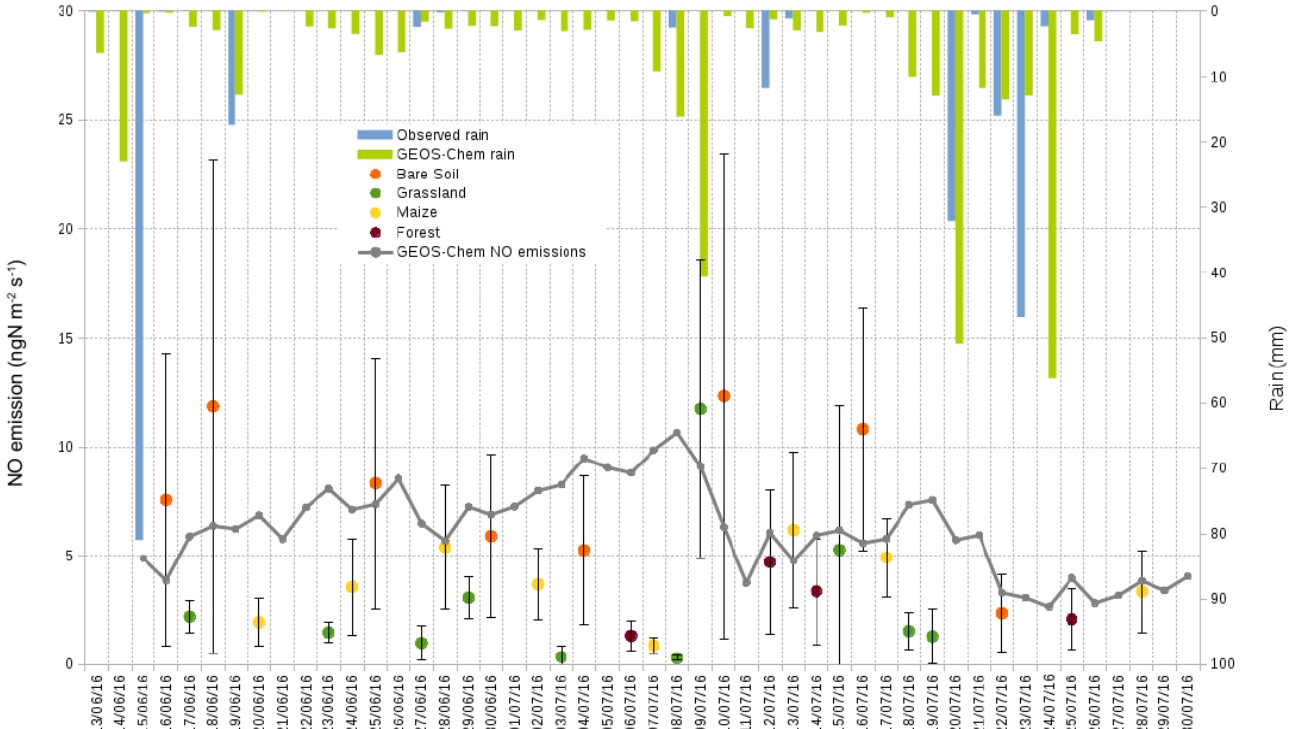

Fig. 6 Nitric oxide emissions in ngN m$^{-2}$ s$^{-1}$ measured over each land cover type (orange dot for bare soil, green for grassland, yellow for the maize field and brown for forest) and simulated with GEOS-Chem, along with rainfall measured and modelled with GEOS-Chem. Soil NO emissions are daily average between 8 a.m. and 6 p.m..

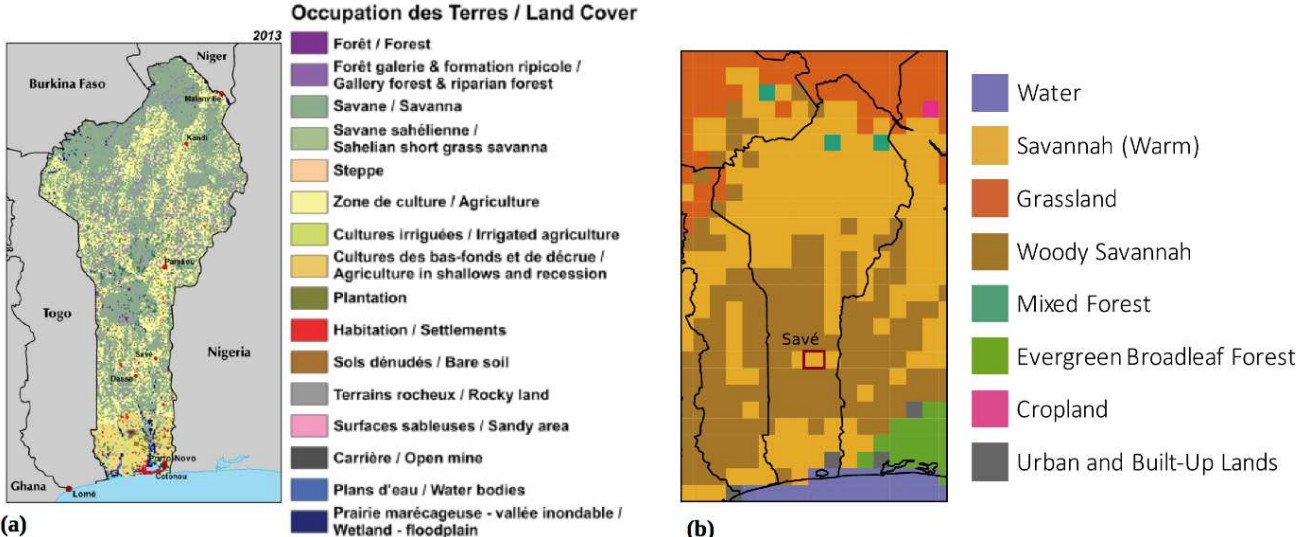

Fig. 7 (a) Land cover map of Benin for 2013 from the US Geographical Survey Atlas: Landscapes of West Africa – A Window on a Changing World (CILSS, 2016) and (b) land cover map of Benin used in the GEOS-Chem simulation.