# Peer review of "Measurements of nitric oxide and ammonia soil fluxes from a wet savanna ecosystem site in West Africa during the DACCIWA field campaign."

_Atmospheric Chemistry and Physics, 2017_

## Referee Comment (RC1) · Anonymous Referee #2 · 16 Apr 2018

In "Measurements of nitric oxide and ammonia soil fluxes from a wet savanna ecosystem site in West Africa during the DACCIWA field campaign", Pacifico and colleagues measured fluxes of NO and ammonia as well as soil physicochemical properties from four different landscapes in the wet savanna of western Africa (Benin). Atmospheric measurements of this specific ecosystem are much rarer than similar analyses done in forests or urban systems, implicating the importance of a study like this. Additionally, measurements from this region of Africa are few, which makes global modeling efforts difficult. To this regard, I would argue that this manuscript is within the scope of

ACP and presents novel data on N-gas fluxes. Additionally, the authors seem to have successfully interpreted their observations within the bounds of their experiment, and briefly asses the broader significance.

While I believe the methodologies associated with atmospheric and gas flux measurements are sound, I am concerned with the soil sampling protocol and subsequent characterization, especially as it relates to mineral-N. It is not clear if the authors took 3-4 replicates for each landscape element each day, or a total of 3-4 replicates over the course of the campaign for each landscape element (lines 243-244); however, I assume it is the later. Because of this, the authors report high levels of variability in organic C and N from 06/07/2016 to 28/07/2016. Soil C should not change (to the extent reported) over the course of 3 days (Bare Soil: 06/07-09/07) and can only be attributed to environmental heterogeneity; however, the authors do somewhat account for this in their analysis by averaging all values over the course of the campaign. Prior to analysis, the authors chose to air-dry soil and store for two months. While I am sure location and resources had much to do with this, air-drying may result in large changes to ammonium concentrations. Additionally, significant changes in the amounts of ammonium can take place over prolonged storage at room temperature, even if soils are dried. It seems that the authors are aware of this issue and attempted to justify their method by citing a meta-analysis of warming experiments on N-cycle activity. (Bai et al. 2013). However, this meta-analysis found that warming and moisture reduction had no significant effect on mineralization (Bai et al, 2013: Table 1), indicating even in dried samples, pools of inorganic-N may change over time. To remedy this, the authors could have compared their ammonium concentrations to similar studies from this region; however, this was not included in the results/discussion. That being said, it is somewhat gratifying that soil emission potentials are in line with a previous study (Massad et al. 2010).

In regards to the tables and figures, the authors should strongly consider merging Figures 2-5. As they currently sit, there is a large amount of redundancy.

---

## Referee Comment (RC2) · Anonymous Referee #1 · 18 Apr 2018

1. General comments:

The manuscript by Pacifico et al. presents a study of nitric oxide (NO) and ammonia (NH3) soil fluxes form four different land cover types within a wet savanna ecosystem in West Africa. The biogeochemical cycles of both NO and NH3 are strongly altered by anthropogenic activity and their surface-atmosphere exchange has important impacts on atmospheric chemistry and air quality. With a strong increase of anthropogenic emissions and only little or no observations in this region of the world, the presented study is an important contribution.

[Figure]

The manuscript is structured well and the research is presented in a scientifically sound way. However, there are some main aspects which are lacking clarification and have to be improved:

The interpretation of presented fluxes largely depends on the quality of the flux measurements. For the determination of fluxes using the closed-dynamic chamber technique the authors use several assumptions without discussing their validity. For example, it cannot be assumed that there are no interactions of NH3 with the chamber walls without adequate test experiments. While I highly acknowledge the fact that measurements in the present study region are challenging, I believe the validity of the used assumptions has to be tested. An accurate flux error assessment is especially necessary for NH3, which is subject to bi-directional exchange, and might explain some of the strong variability of the presented results.

The authors use the measured NO and NH3 fluxes for a stepwise linear multiple regression analysis, upscaling to country-wide soil fluxes, and comparison with soil emission estimates from the GEOS-Chem model. These analyses give valuable information on the importance of soil NO and NH3 exchange and our current knowledge about them. However, while the authors state that a process understanding of the NO and NH3 fluxes is not within the scope of the presented study, in my opinion it is important to understand the underlying processes of the measured fluxes. For example, the estimated emissions from soil characteristics only poorly agree with the measured fluxes in some parts, which indicates that a more detailed process understanding is necessary.

The study focuses on soil fluxes, which is why the authors do not discuss the impact of vegetation on the NO and NH3 fluxes. Especially for NH3, a present canopy may significantly alter the net ecosystem flux and I suggest to add a note including this aspect in the discussion of the manuscript.

2. Specific comments:

L. 50: The study by Oswald et al. (2013) is on soil HONO emissions. Please cite here

the original publication for NO emissions (IPCC or other original source).

L. 159: Active charcoal is mainly suitable for medium to high molecular weight compounds and compounds with low volatility. Hence, I am surprised that the active charcoal was enough to remove all ambient air NH3. Was the quality of zero air source also tested against other methods? If so, please state this in the manuscript.

L. 180: The assumption that the concentration in the chamber is equal to the concentration leaving the chamber to the analyzer is questionable. Due to the low flow rate required for the practical use of the closed-dynamic chamber technique, the residence time within the chamber is substantial (17-18 min). As no active mixing (e.g. with fan) is used, the chamber geometry in relation to the positioning of the ambient air inlet and sample outlet is of importance.

L. 181-182: Especially NH3 is known to be a very sticky molecules and it cannot necessarily be assumed that it does not adsorb to Teflon material. E.g. from online NH3 measurements there is strong evidence that NH3 significantly interacts with the walls of used inlet Teflon tubing, already on a short time scale. The adsorption strength is thought to depend mainly on temperature, presence of NH3 and particulate matter on the Teflon surface, or relative humidity. Likewise, there could be a substantial effect depending on whether the manual chamber was cleaned before each measurement or not. Potential wall effects on NH3 fluxes are an important issue and should be addressed, e.g. by performing a field blank test or adequate laboratory test experiments.

L. 190-193: Both the dilution effect and the detection limit are directly linked to the considered time interval. To my understanding, with longer time intervals the dilution effect increases and the detection limit decreases. As stated in the manuscript, for NO a shorter time interval (120s) was chosen than for NH3 (180-300s) (note here: in Delon et al. (2017), the time intervals for NO and NH3 were the opposite). According to this, the dilution effect is larger for NH3 than for NO, however, the stated detection limit is smaller for an NO than NH3, which should be the opposite. Please correct these

inconsistencies or explain the differences in the revised manuscript.

L. 202-203: Please state if a 1-sigma or 3-sigma detection limit is given here.

L. 227-228: Key for the quality of the closed-dynamic chamber technique is the accurate determination of the initial concentration slope after the chamber installation. For this reason, the authors correctly omit fluxes where the slope is below a threshold correlation coefficient and the measured concentration difference is low. However, especially for NH3, where a R2 threshold value of 0.4 was chosen, the knowledge of the flux error is important for the further interpretation and might explain some of the presented flux variations. Therefore, the authors should include an estimate of the flux error associated with the linear regression and take that into account for the discussion of results.

L. 248-250: This assumptions seems brave if it was not tested with a set of test experiments. Although the microbial activity is reduced due to the dry conditions, there is a chance that NH3 volatilizes with the drying of the soil sample material.

L. 368-439: Presentation of NO and NH3 flux results: The flux at the soil-atmosphere interface is governed next to processes in the soil by the ambient trace gas concentration above the soil surface. The authors report relevant soil properties, while the atmospheric NO and NH3 mixing ratios from the chamber measurement are not reported. As they might significantly impact the magnitude and sign of the fluxes, the authors should include this information in the figures and the manuscript. This is especially important for the interpretation of the NH3 fluxes which are subject to bi-directional exchange and might explain some of the large flux variations observed.

L. 385-387: Why are the underground roots especially important for bare soil and the maize field? Are they more dominant than roots at the grassland and forest site?

L. 431-434: I agree with the authors in addressing the issue of the NH4+ adsorption capacity of soil particles when interpreting the results from the soil measurements.

However, in this context it is also important what method was chosen to determine the soil NH4+. E.g. some common methods use a potassium chloride solution, to extract the soil NH4+. As a consequence, using a strong extraction solution might result in an overestimation of the emission potential.

L. 435-436: The authors bring up the potential of NH3 deposition on water film on vegetation surfaces, although the study focuses on soil emissions. Hence, it is important to mention (e.g. in method section) in case the chamber measurements also incorporated lower growing plant species (e.g. for grassland site) and include that in the interpretation of the results (i.e, stomatal emission potential).

3. Technical comments:

L. 109: Use capital in "Guinean"

L. 122: Use "next to" instead of "next"

L. 124: Use "next to the grassland site" instead of "next to grassland site"

L. 304-306: It is common to use the past tense for reporting on measurement results. Also, total rainfall should be added for completeness.

———————————————

---

## Author Comment (AC1) · 7 Jun 2018

Please see the Supplement for better editing of the same text and figures

The authors greatly thank the reviewer for the interesting and constructive comments on the manuscript. We will try below to answer the questions and propose solutions. The reviewer's question is in italic, while the author's answer is below. The line numbers where the modifications are made correspond to the new version of the manuscript. General comments: The interpretation of presented fluxes largely depends on the quality of the flux measurements. For the determination of fluxes using the closed-dynamic chamber technique the authors use several assumptions without discussing their validity. For example, it cannot be assumed that there are no interactions of NH3 with the chamber walls without adequate test experiments. While I highly acknowledge the fact that measurements in the present study region are challenging, I believe the validity of the used assumptions has to be tested. An accurate flux error assessment is especially necessary for NH3, which is subject to bi-directional exchange, and might explain some of the strong variability of the presented results. The referee is totally right in stating that experimental tests are lacking on ammonia adsorption on Teflon chamber walls, in comparison with other surfaces. However, we have trusted the literature on that specific question. Vaittinen et al. (2013, and references therein), have systematically assessed the adsorption of gas phase NH3 on various surface materials. They have stated that polymers generally adsorb less NH3 than stainless steel surfaces (giving that adsorption is the difference between the inlet and outlet mixing ratio). 12 molecules.cm on PFA. The surface of our chamber is 2700 cm , which gives a value of As an example, in their Table 2, they assess that the adsorption of ammonia is 13.9.10 -2 2 12 concentrationsmeasuredonthefield).Thevolumeofthechamberis12300cm (12.3liters).One 37530.10 molecules adsorbed (for a given concentration of 8.5 ppb, which is in the upper range of 3 23 23 12 -5 3.3.10 molecules compared to 37530.10 gives an amount of 1.14.10 % adsorbed on the surface of the chamber. The surface of the Teflon tubing is $2\pi$*0.22*400 cm2 = 553 cm2, i.e. it may adsorb 553*13.9.1012 = mole of gas occupies 22.4 liters, therefore the chamber contains 0.55 moles of gas, i.e. 3.3.10 molecules. 12 -7 7685.10 We therefore suggest that the flux error, relative to ammonia adsorption on Teflon walls, is negligible. The following text has been added line 178: (Vaittinen et al. (2013, and references therein), have demonstrated that the adsorption of ammonia on Teflon walls is negligible) The authors use the measured NO and NH3 fluxes for a stepwise linear multiple regression analysis, upscaling to country-wide soil fluxes, and comparison with soil emission estimates from the GEOS- Chem model. These analyses give

valuable information on the importance of soil NO and NH3 exchange and our current knowledge about them. However, while the authors state that a process understanding of the NO and NH3 fluxes is not within the scope of the presented study, in my opinion it is important to understand the underlying processes of the measured fluxes. For example, molecules (2.3.10 %). the estimated emissions from soil characteristics only poorly agree with the measured fluxes in some parts, which indicates that a more detailed process understanding is necessary. The authors agree that a close understanding of the underlying processes of NO and NH3 exchanges is important. There is a misunderstanding of what we wanted to state by writing that the understanding of these processes is not in the scope of this study. Indeed, in our study, we could not find any strong correlation between soil moisture and fluxes, or soil temperature and fluxes, because of the spatial set up of the experiment (with 4 types of soil described). However, this set up is useful to represent the spatial variability of fluxes. This may justify why we did not present any plot of flux magnitude vs environmental variable, but rather temporal evolution of fluxes and comparison with modeling results. Line 520, we have removed the sentence "which is not in the scope of this study" The study focuses on soil fluxes, which is why the authors do not discuss the impact of vegetation on the NO and NH3 fluxes. Especially for NH3, a present canopy may significantly alter the net ecosystem flux and I suggest to add a note including this aspect in the discussion of the manuscript. Yes indeed, the role of vegetation in regulating NH3 and NO fluxes from soils is of prime importance. Line 419 we added the following paragraph: The NO flux estimated in this study takes into account the ground flux only. Indeed, the net emission to the atmosphere should take into account the oxidation of NO in NO2 and the eventual re-deposition of NO2 on the vegetation called the Canopy Reduction Factor, assumed to be a linear function of Leaf Area Index (defined for example in Yienger and Levy (1995) and Ganzeveld et al.(2002)). Line 457 we added the following paragraph: Our measurements were conducted without vegetation inside the chambers, but vegetation was present in the fields. It is important to mention that the role of vegetation on NH3 bidirectional fluxes is essential, especially during the wet

season (time of the experiment), when deposition on the vegetation through stomata and cuticles dominate the exchange (during rain events, the cuticular resistance becomes small and cuticular deposition dominates), due to an increase of the deposition velocity of NH3 (consecutive to the humidity response of the surface) and a decrease of the canopy compensation point, sensitive to the surface temperature and the surface wetness (Wichink-Kruit et al., 2007). Specific comments L. 50: The study by Oswald et al. (2013) is on soil HONO emissions. Please cite here the original publication for NO emissions (IPCC or other original source). The reference will be changed by the IPCC reference. L. 159: Active charcoal is mainly suitable for medium to high molecular weight compounds and compounds with low volatility. Hence, I am surprised that the active charcoal was enough to remove all ambient air NH3. Was the quality of zero air source also tested against other methods? If so, please state this in the manuscript.

Line 157, the following explanations have been added: "....and a second one to validate the efficiency of the NH3 converter with a NH3/N2 mixture diluted in pure air (Alphagaz 1, Airliquide). The zero air for NO, NO2 calibration was obtained by filtering ambient air, previously passed on charcoal and desiccant cartridges." L. 180: The assumption that the concentration in the chamber is equal to the concentration leaving the chamber to the analyzer is questionable. Due to the low flow rate required for the practical use of the closed-dynamic chamber technique, the residence time within the chamber is substantial (17- 18 min). As no active mixing (e.g. with fan) is used, the chamber geometry in relation to the positioning of the ambient air inlet and sample outlet is of importance. As described in Delon et al. (2017): The external volume of the chamber was 40 cm $\times$ 20 cm $\times$ 20 cm. The useful volume was 18 $\times$38 $\times$18 cm3 (12.3l or 0.0123m3), due to the thickness of the Teflonwalls. The air inlet is on one side of the chamber (a small vent of 4 mm in diameter provided the pressure equilibrium between the inside and outside of the chamber). The air outlet on the other side (40 cm away) is connected to the analyzer with a 4 m Teflon tube. These specifications are not recalled in the manuscript, we just added: Line 181: the details of the calculation "and chamber design" are given in Delon et al. (2017). L. 181-182: Especially NH3 is known to be a

very sticky molecules and it cannot necessarily be assumed that it does not adsorb to Teflon material. E.g. from online NH3 measurements there is strong evidence that NH3 significantly interacts with the walls of used inlet Teflon tubing, already on a short time scale. The adsorption strength is thought to depend mainly on temperature, presence of NH3 and particulate matter on the Teflon surface, or relative humidity. Likewise, there could be a substantial effect depending on whether the manual chamber was cleaned before each measurement or not. Potential wall effects on NH3 fluxes are an important issue and should be addressed, e.g. by performing a field blank test or adequate laboratory test experiments. This point was mentioned in the general comments, and considering the literature, we propose to neglect the adsorption of NH3 on Teflon walls and Teflon tubing (for particular temperature, humidity and concentration conditions). Of course, the temperature and humidity conditions in our experiment are different, but even if the number of adsorbed molecules increases, it may not bring discredit on our results A sentence has been added in the manuscript line 181. During this experiment, unfortunately no field blank test has been made. Line 173 we added the following text: The Teflon chamber was cleaned at the beginning of each day of measurement, and during the day when the deposition of sand was considered too important in it. Line 221 we added the following text: The measurements have not been corrected from a possible interaction with particulate matter. L. 190-193: Both the dilution effect and the detection limit are directly linked to the considered time interval. To my understanding, with longer time intervals the dilution effect increases and the detection limit decreases. As stated in the manuscript, for NO a shorter time interval (120s) was chosen than for NH3 (180-300s) (note here: in Delon et al. (2017), the time intervals for NO and NH3 were the opposite). According to this, the dilution effect is larger for NH3 than for NO, however, the stated detection limit is smaller for an NO than NH3, which should be the opposite. Please correct these inconsistencies or explain the differences in the revised manuscript. Thanks for mentioning this inconsistency.

Actually, the considered time interval varies in a range between 100 and 300s both for NH3 and NO (this is different from Delon et al., 2017). The dilution effect is calculated

separately for each flux and the correction is 7.7 ($\pm$1.7) % in average for NH3 fluxes, and 6.7 ($\pm$1.6) % in average for NO fluxes. The minimum flux measurable is calculated from the precision of the instrument (and not from the detection limit), which is $\pm$0.4 ppbv f, for a 10s time interval. It does not depend on the time interval used for the calculation of the linear regression. The minimum flux detected by this device would -2 -1 therefore be 0.4 ngN m s for NH3 and for NO. Line 190 the paragraph has been changed into: The linear regression is calculated over a 100 to 300 s time interval after the installation of the chamber on soil for both NO and NH3. Based on the methodology developed in Delon et al. (2017), the dilution effect due to mixing of outside air in the chamber is calculated for each flux separately and is in average 6.7($\pm$1.6) % for NO and 7.7($\pm$1.7) % for NH3. Considering the precision of the analyzer ($\pm$0.4 ppbv), the detection limit is 0.4 ngN m-2 s-1 for NO and NH3 fluxes (different from Delon et al., 2017, but fluxes were anyway superior to this value). L. 202-203: Please state if a 1-sigma or 3-sigma detection limit is given here. A 2-sigma detection limit is given, added line 201 L. 227-228: Key for the quality of the closed-dynamic chamber technique is the accurate determination of the initial concentration slope after the chamber installation. For this reason, the authors correctly omit fluxes where the slope is below a threshold correlation coefficient and the measured concentration difference is low. However, especially for NH3, where a R2 threshold value of 0.4 was chosen, the knowledge of the flux error is important for the further interpretation and might explain some of the presented flux variations. Therefore, the authors should include an estimate of the flux error associated with the linear regression and take that into account for the discussion of results. Line 226, the following point was added: A flux error was estimated, by calculating the dispersion of points around the linear regression line used to calculate the slope. According to this method, the dispersion for NO flux calculation is comprised between 5 and 12%, and the dispersion for NH3 flux calculation is comprised between 15 and 20%.. L. 248-250: This assumption seems brave if it was not tested with a set of test experiments. Although the microbial activity is reduced due to the dry conditions, there is a chance that NH3 volatilizes with the drying of the soil sample material. The

authors are aware of this particular problem of NH3 volatilization. The analyses have been made as soon as possible after the field campaign. A direct analysis was not possible, due to missing infrastructure. Freezing the samples would have been the best solution, but considering the difficulty of organizing the campaign in a place where the minimum material was installed, we could not afford to bring a freezer. Line 255, new references have been added in the text: Some authors have also published results of ammonium concentrations measured in soils that were dried in ambient air (Bai et al., 2010, Dick et al., 2006, Cassity-Duffrey et al., 2015).

Very few values of ammonium concentrations in African soils are available in the literature, it is therefore very difficult to compare. Line 360, the following references have been added: + -1 Dick et al. (2006) have found NH4 concentrations between 2 and 8 mgN.kg in Senegalese soils, which is very close from our results. Vanlauwe et al. (2002) have found values between 0.8 and 1.4 -1 mgN.kg in West African moist savanna soils (in Togo and Nigeria).. L. 368-439: Presentation of NO and NH3 flux results: The flux at the soil-atmosphere interface is governed next to processes in the soil by the ambient trace gas concentration above the soil surface. The authors report relevant soil properties, while the atmospheric NO and NH3 mixing ratios from the chamber measurement are not reported. As they might significantly impact the magnitude and sign of the fluxes, the authors should include this information in the figures and the manuscript. This is especially important for the interpretation of the NH3 fluxes which are subject to bi- directional exchange and might explain some of the large flux variations observed. Figure 2 to 5 have been modified and concentrations have been added (see new figures at the end of this document) In section 3.4 line 391 we have added the following paragraph: Daily means of NO concentrations are measured close to the soil (0.1m, half height of the chamber) and reported in fig. 2 to 5. Daily means of NO concentration vary from 1.28 to 5.40 ppb for all sites. The average concentration during the whole campaign on all sites is 2.70 $\pm$ 1.03 ppb. Average concentration on bare soil is 2.97 $\pm$ 1.49 ppb, it is 2.57 $\pm$ 0.96 ppb on grassland, 2.55 $\pm$ 0.83 on maize, and 2.76 $\pm$ 0.65 ppb on forest soils. The concentrations are quasi equivalent for all

sites. As these concentrations are low, they do not lead to NO deposition on the soil, the NO flux keeps being positive. Indeed, NO deposition has already been measured but only in the case of high NO concentrations (>60ppb, Laville et al., 2011). In section 3.5 line 437 we have added the following paragraph: As for NO concentrations, NH3 concentrations are reported in fig. 2 to 5. Daily means of NH3 concentration vary from 0 to 12.46 ppb for all sites, and the average concentration is 4.42 $\pm$ 3.23 ppb during the whole campaign. Average NH3 concentration for bare soils is 6.28 $\pm$ 3.90 ppb, it is 3.28 $\pm$ 1.79 ppb for grassland, 4.36 $\pm$ 3.99 for maize field, and 3.68 $\pm$ 2.13 ppb for forest. The largest deposition fluxes are found on bare soils, where the largest concentrations are measured. L. 385-387: Why are the underground roots especially important for bare soil and the maize field? Are they more dominant than roots at the grassland and forest site? Line 397 the paragraph has been modified: The spatial variability of NO fluxes is high, for bare soil, forest and the maize field where underground roots (not visible at the surface) are heterogeneously distributed. These roots are likely to influence the ammonium content of the soil and the subsequent NO flux measurement. Standard th th deviation is generally smaller for grassland (except for two days, July 9 and 13 ) , where the vegetation (and the root distribution) is more homogeneous. The repartition of the vegetation is more homogeneous in grassland. Roots are present everywhere, and their presence is less variable in grassland plots, whereas bare soil and maize plots present a larger heterogeneity. L. 431-434: I agree with the authors in addressing the issue of the NH4+ adsorption capacity of soil particles when interpreting the results from the soil measurements. However, in this context it is also important what method was chosen to determine the soil NH4+. E.g. some

common methods use a potassium chloride solution, to extract the soil NH4+. As a consequence, using a strong extraction solution might result in an overestimation of the emission potential. Material and Method section mentions that mineral and organic nitrogen are determined following norm NF ISO 13878. There is an error in the text. NF ISO 13878 is used for Total N. This has been corrected. Line 263 the following text has been added: Mineral nitrogen is determined following an internal method MT-

AZM adapted from norm NF ISO 14256-2. This method uses a potassium chloride solution and is COFRAC certified. PS: the concentration is unknown because the lab does not want to communicate on it. L. 435-436: The authors bring up the potential of NH3 deposition on water film on vegetation surfaces, although the study focuses on soil emissions. Hence, it is important to mention (e.g. in method section) in case the chamber measurements also incorporated lower growing plant species (e.g. for grassland site) and include that in the interpretation of the results (i.e, stomatal emission potential). All the measurements were made on direct soil without vegetation, even in the grassland field (the grass was too high to be included in the chamber). However, some short stems or leaves sometimes subsist within the chamber and were not removed to avoid any disturbance of the soil condition, but does not justify to take into account stomatal emission potential (because in too low quantity). The hypothesis of a water film on the vegetation is removed from the sentence; the hypothesis of a water film on the soil surface is left. Technical comments: corrections have been done. The following references were added: Bai Junhong, Haifeng Gao, Wei Deng, Zhifeng Yang, Baoshan Cui, Rong Xiao, Nitrification potential of marsh soils from two natural saline–alkaline wetlands, Biol Fertil Soils (2010) 46:525–529. Cassity-Duffey Kate, Miguel Cabrera, John Rema, Ammonia Volatilization from Broiler Litter: Effect of Soil Water Content and Humidity, Soil Sci. Soc. Am. J. 79:543–550, 2014. Dick Jan, Ute Skiba, Robert Munro and Douglas Deans, Effect of N-fixing and non N-fixing trees and crops on NO and N2O emissions from Senegalese soils, Journal of Biogeography (J. Biogeogr.) (2006) 33, 416–423. Ganzeveld L.N., J. Lelieveld, F. J. Dentener, M. C. Krol, A. J. Bouwman, and G.-J. Roelofs, Global soil-biogenic NOx emissions and the role of canopy processes, JOURNAL OF GEOPHYSICAL RESEARCH, VOL. 107, NO. D16, 10.1029/2001JD001289, 2002. Vaittinen O., M. Metsa ÌLla, S. Persijn, M. Vainio, L. Halonen, Adsorption of ammonia on treated stainless steel and polymer surfaces, Appl. Phys. B, DOI 10.1007/s00340-013-5590-3, 2013. Vanlauwe B., J. Diels, O. Lyasse, K. Aihou, E.N.O. Iwuafor, N. Sanginga, R. Merckx & J. Deckers, Fertility status of soils of the derived savanna and northern guinea savanna and response to major

plant nutrients, as influenced by soil type and land use management, Nutrient Cycling in Agroecosystems 62: 139–150, 2002. Wichink Kruit, R. J., van Pul, W. A. J., Otjes, R. P., Hofschreuder, P., Jacobs, A. F. G., and Holtslag, A. A. M (2007), Ammonia fluxes and derived canopy compensation points over nonfertilized agricultural grassland in the Netherlands using the new gradient ammonia – high accuracy – monitor (GRAHAM), Atmos. Environ., 41, 1275–1287. Yienger, J. J., and H. Levy II, Global inventory of soil-biogenic NOx emissions, J. Geophys. Res., 100, 11,447– 11,464, 1995.

Please also note the supplement to this comment:
https://www.atmos-chem-phys-discuss.net/acp-2017-1198/acp-2017-1198-AC1-supplement.pdf

—————————————————————

---

## Author Comment (AC2) · 7 Jun 2018

The authors greatly thank the reviewer for the interesting and constructive comments on the manuscript. We will try below to answer the questions and propose solutions. The reviewer's question is in italic, while the author's answer is below. Line numbers where modifications are made are relative to the new version of the manuscript. General comments: While I believe the methodologies associated with atmospheric and gas flux measurements are sound, I am concerned with the soil sampling protocol and subsequent characterization, especially as it relates to mineral-N. It is not clear if the

authors took 3-4 replicates for each landscape element each day, or a total of 3-4 replicates over the course of the campaign for each landscape element (lines 243-244). The authors took a total of 3-4 replicates over the course of the campaign. On one particular th th day, samples were collected at the 4 different ecosystems. For example, on July 19 and 28 , samples were collected in bare soil, grassland, forest and maize field, whereas the forest site th th was not sampled on July 9 , and maize field was not sampled on July 6 . th th Moreover, the first collection date for grassland should be July 6 (instead of 7 ). This will be corrected. Line 246, the sentence has been modified in Samples were collected at the four different land cover types, three to four times during the campaign Soil C should not change (to the extent reported) over the course of 3 days (Bare Soil: 06/07- 09/07) and can only be attributed to environmental heterogeneity Actually samples were collected in the same area, but not exactly at the same place (another hole was dug, close to the first one). Therefore, the spatial heterogeneity of samples may explain de different results of organic carbon. While I am sure location and resources had much to do with this, air-drying may result in large changes to ammonium concentrations. Additionally, significant changes in the amounts of ammonium can take place over prolonged storage at room temperature, even if soils are dried. It seems that the authors are aware of this issue and attempted to justify their method by citing a meta-analysis of warming experiments on N-cycle activity. (Bai et al. 2013). However, this meta-analysis found that warming and moisture reduction had no significant effect on mineralization (Bai et al, 2013: Table 1), indicating even in dried samples, pools of inorganic-N may change over time. To remedy this, the authors could have compared their ammonium concentrations to similar studies from this region; however, this was not included in the results/discussion. The authors are aware of this particular problem of NH3 volatilization. The analyses have been made as soon as possible after the field campaign. A direct analysis was not possible, due to missing infrastructure. Freezing the samples would have been the best solution, but considering the difficulty of organizing the campaign in a place where the minimum material was installed, we could not afford to bring a freezer. Some authors have also published

results of ammonium concentrations measured in soils that were dried in ambient air (Bai et al., 2010, Dick et al., 2006, Cassity-Duffrey et al., 2015).

Moreover, very few values are available in the literature, it is therefore very difficult to compare. The results presented in section 3.3 are consistent with Massad et al. (2010) which + concentrations between 2 and 8 mgN.kg results. Vanlauwe et al. (2002) have found values between 0.8 and 1.4 mgN.kg in West African moist savanna soils (in Togo and Nigeria). Line 255, new references have been added in the text: Some authors have also published results of ammonium concentrations measured in soils that were dried in ambient air (Bai et al., 2010, Dick et al., 2006, Cassity-Duffrey et al., 2015). may provide a certain confidence in analysis. Dick et al. (2006) have also found NH4 -1 Line 360, the following references have been added: + -1 in Senegalese soils, which is very close from our -1 Dick et al. (2006) have found NH4 concentrations between 2 and 8 mgN.kg in Senegalese soils, which is very close from our results. Vanlauwe et al. (2002) have found values between 0.8 and 1.4 mgN.kg -1 in West African moist savanna soils (in Togo and Nigeria).. In regards to the tables and figures, the authors should strongly consider merging Figures 2-5. As they currently sit, there is a large amount of redundancy. As reviewer #1 asked to add NO and NH3 concentrations in the figures, we chose not to merge figures 2 to 5, it would have been impossible to read. References Bai Junhong, Haifeng Gao, Wei Deng, Zhifeng Yang, Baoshan Cui, Rong Xiao, Nitrification potential of marsh soils from two natural saline–alkaline wetlands, Biol Fertil Soils (2010) 46:525–529. Cassity-Duffey Kate, Miguel Cabrera, John Rema, Ammonia Volatilization from Broiler Litter: Effect of Soil Water Content and Humidity, Soil Sci. Soc. Am. J. 79:543–550, 2014. Dick Jan, Ute Skiba, Robert Munro and Douglas Deans, Effect of N-fixing and non N- fixing trees and crops on NO and N2O emissions from Senegalese soils, Journal of Biogeography (J. Biogeogr.) (2006) 33, 416–423. Vanlauwe B., J. Diels, O. Lyasse, K. Aihou, E.N.O. Iwuafor, N. Sanginga, R. Merckx & J. Deckers, Fertility status of soils of the derived savanna and northern guinea savanna and response to major plant nutrients, as influenced by soil type and land use management, Nutrient Cycling in Agroecosystems 62: 139–150,

2002.

Please also note the supplement to this comment:
https://www.atmos-chem-phys-discuss.net/acp-2017-1198/acp-2017-1198-AC2-
supplement.pdf
* * *

---

## Author Response (AR1)

**Response to anonymous referee #1**

The authors greatly thank the reviewer for the interesting and constructive comments on the manuscript. We will try below to answer the questions and propose solutions. The reviewer's question is in italic, while the author's answer is below.

*General comments:*

*The interpretation of presented fluxes largely depends on the quality of the flux measurements. For the determination of fluxes using the closed-dynamic chamber technique the authors use several assumptions without discussing their validity. For example, it cannot be assumed that there are no interactions of NH3 with the chamber walls without adequate test experiments. While I highly acknowledge the fact that measurements in the present study region are challenging, I believe the validity of the used assumptions has to be tested. An accurate flux error assessment is especially necessary for NH3, which is subject to bi-directional exchange, and might explain some of the strong variability of the presented results.*

The referee is totally right in stating that experimental tests are lacking on ammonia adsorption on Teflon chamber walls, in comparison with other surfaces. However, we have trusted the literature on that specific question.
Vaittinen et al. (2013, and references therein), have systematically assessed the adsorption of gas phase $NH_3$ on various surface materials. They have stated that polymers generally adsorb less $NH_3$ than stainless steel surfaces (giving that adsorption is the difference between the inlet and outlet mixing ratio).
As an example, in their Table 2, they assess that the adsorption of ammonia is $13.9.10^{12}$ molecules.$cm^{-2}$ on PFA. The surface of our chamber is 2700 $cm^2$, which gives a value of $37530.10^{12}$ molecules adsorbed (for a given concentration of 8.5 ppb, which is in the upper range of concentrations measured on the field). The volume of the chamber is 12300 $cm^3$ (12.3 liters). One mole of gas occupies 22.4 liters, therefore the chamber contains 0.55 moles of gas, i.e. $3.3.10^{23}$ molecules.
$3.3.10^{23}$ molecules compared to $37530.10^{12}$ gives an amount of $1.14.10^{-5}$ % adsorbed on the surface of the chamber.
The surface of the Teflon tubing is $2\pi*0.22*400$ $cm^2 = 553$ $cm^2$, i.e. it may adsorb $553*13.9.10^{12} = 7685.10^{12}$ molecules ($2.3.10^{-7}$ %).
We therefore suggest that the flux error, relative to ammonia adsorption on Teflon walls, is negligible.

**The following text has been added line 181:**
No deposition occurs onto the Teflon walls chamber, as Vaittinen et al. (2013, and references therein) have demonstrated that the adsorption of ammonia on Teflon walls is negligible.

*The authors use the measured NO and NH3 fluxes for a stepwise linear multiple regression analysis, upscaling to country-wide soil fluxes, and comparison with soil emission estimates from the GEOS-Chem model. These analyses give valuable information on the importance of soil NO and NH3 exchange and our current knowledge about them. However, while the authors state that a process understanding of the NO and NH3 fluxes is not within the scope of the presented study, in my opinion it is important to understand the underlying processes of the measured fluxes. For example, the estimated emissions from soil characteristics only poorly agree with the measured fluxes in some parts, which indicates that a more detailed process understanding is necessary.*

The authors agree that a close understanding of the underlying processes of NO and NH$_3$ exchanges is important. There is a misunderstanding of what we wanted to state by writing that the understanding of these processes is not in the scope of this study. Indeed, in our study, we could not find any strong correlation between soil moisture and fluxes, or soil temperature and fluxes, because of the spatial set up of the experiment (with 4 types of soil described). However, this set up is useful to represent the spatial variability of fluxes. This may justify why we did not present any plot of flux magnitude vs environmental variable, but rather temporal evolution of fluxes and comparison with modeling results.

**Line 499,** we have removed the sentence "which is not in the scope of this study"

*The study focuses on soil fluxes, which is why the authors do not discuss the impact of vegetation on the NO and NH3 fluxes. Especially for NH3, a present canopy may significantly alter the net ecosystem flux and I suggest to add a note including this aspect in the discussion of the manuscript.*

Yes indeed, the role of vegetation in regulating NH$_3$ and NO fluxes from soils is of prime importance.

**Line 409** we added the following paragraph:
The NO flux estimated in this study does not consider the impact of vegetation on the net ecosystem flux, as we focus on soil fluxes only. However, the net emission to the atmosphere should take into account the oxidation of NO to NO2 and the eventual re-deposition of NO2 on the vegetation, i.e. what is called Canopy Reduction Factor and is assumed to be a linear function of the Leaf Area Index (e.g. Yienger and Levy, 1995, and Ganzeveld et al., 2002)).

**Line 446** we added the following paragraph:
Our measurements were conducted without vegetation inside the chambers, but vegetation was present in the fields. It is important to mention that the role of vegetation on NH3 bidirectional fluxes is essential, especially during the wet season (time of the experiment), when deposition on the vegetation through stomata and cuticles dominate the exchange (during rain events, the cuticular resistance becomes small and cuticular deposition dominates), due to an increase of the deposition velocity of NH3 (consecutive to the humidity response of the surface) and a decrease of the canopy compensation point, sensitive to the surface temperature and the surface wetness (Wichink-Kruit et al., 2007).

**Specific comments**

*L. 50: The study by Oswald et al. (2013) is on soil HONO emissions. Please cite here the original publication for NO emissions (IPCC or other original source).*

The reference has been changed by the IPCC reference.

*L. 159: Active charcoal is mainly suitable for medium to high molecular weight compounds and compounds with low volatility. Hence, I am surprised that the active charcoal was enough to remove all ambient air NH3. Was the quality of zero air source also tested against other methods? If so, please state this in the manuscript.*

**Line 159,** the following explanations have been added:

"….and a second one to validate the efficiency of the $NH_3$ converter with a $NH_3/N_2$ mixture diluted in pure air (Alphagaz 1, Airliquide). The zero air for NO, $NO_2$ calibration was obtained by filtering ambient air, previously passed on charcoal and desiccant cartridges."

*L. 180: The assumption that the concentration in the chamber is equal to the concentration leaving the chamber to the analyzer is questionable. Due to the low flow rate required for the practical use of the closed-dynamic chamber technique, the residence time within the chamber is substantial (17-18 min). As no active mixing (e.g. with fan) is used, the chamber geometry in relation to the positioning of the ambient air inlet and sample outlet is of importance.*

As described in Delon et al. (2017): The external volume of the chamber was 40 cm × 20 cm × 20 cm. The useful volume was 18 ×38 ×18 cm3 (12.3l or 0.0123m3), due to the thickness of the Teflonwalls. The air inlet is on one side of the chamber (a small vent of 4 mm in diameter provided the pressure equilibrium between the inside and outside of the chamber). The air outlet on the other side (40 cm away) is connected to the analyzer with a 4 m Teflon tube.
These specifications are not recalled in the manuscript, we just added:
**Line 181**: the details of the calculation "and the chamber design" are given in Delon et al. (2017).

*L. 181-182: Especially NH3 is known to be a very sticky molecules and it cannot necessarily be assumed that it does not adsorb to Teflon material. E.g. from online NH3 measurements there is strong evidence that NH3 significantly interacts with the walls of used inlet Teflon tubing, already on a short time scale. The adsorption strength is thought to depend mainly on temperature, presence of NH3 and particulate matter on the Teflon surface, or relative humidity. Likewise, there could be a substantial effect depending on whether the manual chamber was cleaned before each measurement or not. Potential wall effects on NH3 fluxes are an important issue and should be addressed, e.g. by performing a field blank test or adequate laboratory test experiments.*

This point was mentioned in the general comments, and considering the literature, we propose to neglect the adsorption of $NH_3$ on Teflon walls and Teflon tubing (for particular temperature, humidity and concentration conditions). Of course, the temperature and humidity conditions in our experiment are different, but even if the number of adsorbed molecules increases, it may not bring discredit on our results A sentence has been added in the manuscript **line 181**. During this experiment, unfortunately no field blank test has been made.
**Line 173** we added the following text:
The Teflon chamber was cleaned at the beginning of each day of measurement, and during the day when the deposition of sand could potentially interfere with the measurements.
**Line 225** we added the following text:
The measurements have not been corrected from a possible interaction with particulate matter.

*L. 190-193: Both the dilution effect and the detection limit are directly linked to the considered time interval. To my understanding, with longer time intervals the dilution effect increases and the detection limit decreases. As stated in the manuscript, for NO a shorter time interval (120s) was chosen than for NH3 (180-300s) (note here: in Delon et al. (2017), the time intervals for NO and NH3 were the opposite). According to this, the dilution effect is larger for NH3 than for NO, however, the stated detection limit is smaller for an NO than NH3, which should be the opposite. Please correct these inconsistencies or explain the differences in the revised manuscript.*

Thanks for mentioning this inconsistency.
Actually, the considered time interval varies in a range between 100 and 300s both for $NH_3$ and NO (this is different from Delon et al., 2017). The dilution effect is calculated separately for each flux

and the correction is 7.7 (±1.7) % in average for $NH_3$ fluxes, and 6.7 (±1.6) % in average for NO fluxes.

The minimum flux measurable is calculated from the precision of the instrument (and not from the detection limit), which is ±0.4 ppbv f, for a 10s time interval. It does not depend on the time interval used for the calculation of the linear regression. The minimum flux detected by this device would therefore be 0.4 ngN m$^{-2}$ s$^{-1}$ for NH3 and for NO.

**Line 190** the paragraph has been changed into:
The linear regression is calculated over a 100 to 300 s time interval after the installation of the chamber on soil for both NO and NH3. Based on the methodology developed in Delon et al. (2017), the dilution effect due to mixing of outside air in the chamber is calculated for each flux separately and is in average 6.7(±1.6) % for NO and 7.7(±1.7) % for NH3. Considering the precision of the analyzer (±0.4 ppbv), the detection limit is 0.4 ngN m-2 s-1 for NO and NH3 fluxes

This is different from Delon et al., 2017, but fluxes were anyway superior to this value.

*L. 202-203: Please state if a 1-sigma or 3-sigma detection limit is given here.*

A 2-sigma detection limit is given, added **line 203**

*L. 227-228: Key for the quality of the closed-dynamic chamber technique is the accurate determination of the initial concentration slope after the chamber installation. For this reason, the authors correctly omit fluxes where the slope is below a threshold correlation coefficient and the measured concentration difference is low. However, especially for NH3, where a R2 threshold value of 0.4 was chosen, the knowledge of the flux error is important for the further interpretation and might explain some of the presented flux variations. Therefore, the authors should include an estimate of the flux error associated with the linear regression and take that into account for the discussion of results.*

**Line 229**, the following point was added:
A flux error was estimated by calculating the dispersion of points around the linear regression's slope. According to this method, the dispersion for NO flux calculation is comprised between 5 and 12%, and the dispersion for $NH_3$ flux calculation is comprised between 15 and 20%..

*L. 248-250: This assumption seems brave if it was not tested with a set of test experiments. Although the microbial activity is reduced due to the dry conditions, there is a chance that NH3 volatilizes with the drying of the soil sample material.*

The authors are aware of this particular problem of $NH_3$ volatilization. The analyses have been made as soon as possible after the field campaign. A direct analysis was not possible, due to missing infrastructure. Freezing the samples would have been the best solution, but considering the difficulty of organizing the campaign in a place where the minimum material was installed, we could not afford to bring a freezer.

**Line 257**, new references have been added in the text:
Some authors have also published results of ammonium concentrations measured in soils that were dried in ambient air (Bai et al., 2010, Dick et al., 2006, Cassity-Duffrey et al., 2015).

Very few values of ammonium concentrations in African soils are available in the literature, it is therefore very difficult to compare.

**Line 360**, the following references have been added:

Dick et al. (2006) have found $NH_4^+$ concentrations between 2 and 8 $mgN.kg^{-1}$ in Senegalese soils, which is very close from our results. Vanlauwe et al. (2002) have found values between 0.8 and 1.4 $mgN.kg^{-1}$ in West African moist savanna soils (in Togo and Nigeria)..

*L. 368-439: Presentation of NO and NH3 flux results: The flux at the soil-atmosphere interface is governed next to processes in the soil by the ambient trace gas concentration above the soil surface. The authors report relevant soil properties, while the atmospheric NO and NH3 mixing ratios from the chamber measurement are not reported. As they might significantly impact the magnitude and sign of the fluxes, the authors should include this information in the figures and the manuscript. This is especially important for the interpretation of the NH3 fluxes which are subject to bi-directional exchange and might explain some of the large flux variations observed.*

**Figure 2 to 5** have been modified and concentrations have been added (see new figures at the end of this document)

**In section 3.4 line 391** we have added the following paragraph:

Daily means of NO concentrations are measured close to the soil (0.1m, half height of the chamber) and reported in fig. 2 to 5. Daily means of NO concentration vary from 1.28 to 5.40 ppb for all sites. The average concentration during the whole campaign on all sites is 2.70 ± 1.03 ppb. Average NO concentration is 2.97 ± 1.49 ppb on bare soil, 2.57 ± 0.96 ppb on grassland, 2.55 ± 0.83 on maize, and 2.76 ± 0.65 ppb on forest soil. The concentrations are quasi equivalent for all sites. As these concentrations are low, they do not lead to NO deposition on soil and the NO flux stays positive. In fact, NO deposition has been measured in other studies only in the case of high NO concentrations (>60ppb, Laville et al., 2011).

**In section 3.5 line 442** we have added the following paragraph:

As for NO concentrations, $NH_3$ concentrations are reported in fig. 2 to 5. Daily means of $NH_3$ concentration vary from 0 to 12.46 ppb for all sites, and the average concentration is 4.42 ± 3.23 ppb during the whole campaign. Average $NH_3$ concentration is 6.28 ± 3.90 ppb for bare soils, 3.28 ± 1.79 ppb for grassland, 4.36 ± 3.99 for the maize field, and 3.68 ± 2.13 440 ppb for forest.. The largest deposition fluxes are found on bare soils, where the largest concentrations are measured.

*L. 385-387: Why are the underground roots especially important for bare soil and the maize field? Are they more dominant than roots at the grassland and forest site?*

**Line 403** the paragraph has been modified:

The spatial variability of NO fluxes is high, for bare soil, forest and the maize field where underground roots (not visible at the surface) are heterogeneously distributed. These roots are likely to influence the ammonium content of the soil and the subsequent NO flux measurement. Standard deviation is generally smaller for grassland (except for two days, July 9[th] and 13[th]) , where the vegetation (and the root distribution) is more homogeneous.

*L. 431-434: I agree with the authors in addressing the issue of the NH4+ adsorption capacity of soil particles when interpreting the results from the soil measurements. However, in this context it is also important what method was chosen to determine the soil NH4+. E.g. some common methods use a potassium chloride solution, to extract the soil NH4+. As a consequence, using a strong extraction solution might result in an overestimation of the emission potential.*

Material and Method section mentions that mineral and organic nitrogen are determined following norm NF ISO 13878.

There is an error in the text. NF ISO 13878 is used for Total N. This has been corrected.

**Line 263** the following text has been added:

Mineral nitrogen is determined following an internal method MT-AZM adapted from norm NF ISO 14256-2. This method uses a potassium chloride solution and is COFRAC certified.

PS: the concentration is unknown because the lab does not want to communicate on it.

*L. 435-436: The authors bring up the potential of NH3 deposition on water film on vegetation surfaces, although the study focuses on soil emissions. Hence, it is important to mention (e.g. in method section) in case the chamber measurements also incorporated lower growing plant species (e.g. for grassland site) and include that in the interpretation of the results (i.e, stomatal emission potential).*

All the measurements were made on direct soil without vegetation, even in the grassland field (the grass was too high to be included in the chamber). However, some short stems or leaves sometimes subsist within the chamber and were not removed to avoid any disturbance of the soil condition, but does not justify to take into account stomatal emission potential (because in too low quantity).

The hypothesis of a water film on the vegetation is removed from the sentence; the hypothesis of a water film on the soil surface is left**. (line 466).**

***Technical comments: corrections have been done.***

**The following references were added:**

Bai Junhong, Haifeng Gao, Wei Deng, Zhifeng Yang, Baoshan Cui, Rong Xiao, Nitrification potential of marsh soils from two natural saline–alkaline wetlands, Biol Fertil Soils (2010) 46:525–529.

Cassity-Duffey Kate, Miguel Cabrera, John Rema, Ammonia Volatilization from Broiler Litter: Effect of Soil Water Content and Humidity, Soil Sci. Soc. Am. J. 79:543–550, 2014.

Dick Jan, Ute Skiba, Robert Munro and Douglas Deans, Effect of N-fixing and non N-fixing trees and crops on NO and N2O emissions from Senegalese soils, Journal of Biogeography (J. Biogeogr.) (2006) 33, 416–423.

Ganzeveld L.N., J. Lelieveld, F. J. Dentener, M. C. Krol, A. J. Bouwman, and G.-J. Roelofs, Global soil-biogenic NOx emissions and the role of canopy processes, JOURNAL OF GEOPHYSICAL RESEARCH, VOL. 107, NO. D16, 10.1029/2001JD001289, 2002.

Vaittinen O., M. Metsa̍la, S. Persijn, M. Vainio, L. Halonen, Adsorption of ammonia on treated stainless steel and polymer surfaces, Appl. Phys. B, DOI 10.1007/s00340-013-5590-3, 2013.

Vanlauwe B., J. Diels, O. Lyasse, K. Aihou, E.N.O. Iwuafor, N. Sanginga, R. Merckx & J. Deckers, Fertility status of soils of the derived savanna and northern guinea savanna and response to major plant nutrients, as influenced by soil type and land use management, Nutrient Cycling in Agroecosystems 62: 139–150, 2002.

Wichink Kruit, R. J., van Pul, W. A. J., Otjes, R. P., Hofschreuder, P., Jacobs, A. F. G., and Holtslag, A. A. M (2007), Ammonia fluxes and derived canopy compensation points over non-fertilized agricultural grassland in the Netherlands using the new gradient ammonia – high accuracy – monitor (GRAHAM), Atmos. Environ., 41, 1275–1287.

Yienger, J. J., and H. Levy II, Global inventory of soil-biogenic NOx emissions, J. Geophys. Res., 100, 11,447– 11,464, 1995.

[Figure]

Fig. 2 Upper panel: Daily total precipitation (mm), daily mean soil moisture at 5 cm (%) measured by the Karlsruhe Institute of Technology (KIT), daily mean soil moisture averaged between 0 and 30 cm measured by the Université Paul Sabatier (UPS) instrumentation; Middle panel: daily mean NO and $NH_3$ fluxes in ngN m$^{-2}$ s$^{-1}$ measured at the bare soil site; Lower panel: daily mean NO and $NH_3$ concentrations in ppb measured at the bare soil site. Vertical bars show the standard deviation from individual fluxes and concentrations.

[Figure]

Fig. 3 Upper panel: Daily total precipitation (mm), daily mean soil moisture at 5 cm (%) measured by the Karlsruhe Institute of Technology (KIT), daily mean soil moisture averaged between 0 and 30 cm measured by the Université Paul Sabatier (UPS) instrumentation; Middle panel: daily mean NO and $NH_3$ fluxes in ngN m$^{-2}$ s$^{-1}$ measured at the grassland site; Lower panel: daily mean NO and $NH_3$ concentrations in ppb measured at the grassland site. Vertical bars show the standard deviation from individual fluxes and concentrations.

[Figure]

Fig. 4 Upper panel: Daily total precipitation (mm), daily mean soil moisture at 5 cm (%) measured by the Karlsruhe Institute of Technology (KIT), daily mean soil moisture averaged between 0 and 30 cm measured by the Université Paul Sabatier (UPS) instrumentation; Middle panel: daily mean NO and $NH_3$ fluxes in ngN m$^{-2}$ s$^{-1}$ measured at the maize field site; Lower panel: daily mean NO and $NH_3$ concentrations in ppb measured at the maize field site. Vertical bars show the standard deviation from individual fluxes and concentrations.

[Figure]

Fig. 5 Upper panel: Daily total precipitation (mm), daily mean soil moisture at 5 cm (%) measured by the Karlsruhe Institute of Technology (KIT), daily mean soil moisture averaged between 0 and 30 cm measured by the Université Paul Sabatier (UPS) instrumentation; Middle panel: daily mean NO and NH$_3$ fluxes in ngN m$^{-2}$ s$^{-1}$ measured at the forest site; Lower panel: daily mean NO and NH$_3$ concentrations in ppb measured at the forest site. Vertical bars show the standard deviation from individual fluxes and concentrations.

**Response to anonymous referee #2**

The authors greatly thank the reviewer for the interesting and constructive comments on the manuscript. We will try below to answer the questions and propose solutions. The reviewer's question is in italic, while the author's answer is below. Line numbers where modifications are made are relative to the new version of the manuscript.

***General comments:***
*While I believe the methodologies associated with atmospheric and gas flux measurements are sound, I am concerned with the soil sampling protocol and subsequent characterization, especially as it relates to mineral-N. It is not clear if the authors took 3-4 replicates for each landscape element each day, or a total of 3-4 replicates over the course of the campaign for each landscape element (lines 243-244).*

The authors took a total of 3-4 replicates over the course of the campaign. On one particular day, samples were collected at the 4 different ecosystems. For example, on July 19[th] and 28[th], samples were collected in bare soil, grassland, forest and maize field, whereas the forest site was not sampled on July 9[th], and maize field was not sampled on July 6[th].
Moreover, the first collection date for grassland should be July 6[th] (instead of 7[th]). This will be corrected.
**Line 246**, the sentence has been modified in
Samples were collected at the four different land cover types, three to four times during the campaign

*Soil C should not change (to the extent reported) over the course of 3 days (Bare Soil: 06/07-09/07) and can only be attributed to environmental heterogeneity*

Actually samples were collected in the same area, but not exactly at the same place (another hole was dug, close to the first one). Therefore, the spatial heterogeneity of samples may explain de different results of organic carbon.

*While I am sure location and resources had much to do with this, air-drying may result in large changes to ammonium concentrations. Additionally, significant changes in the amounts of ammonium can take place over prolonged storage at room temperature, even if soils are dried. It seems that the authors are aware of this issue and attempted to justify their method by citing a meta-analysis of warming experiments on N-cycle activity. (Bai et al. 2013). However, this meta-analysis found that warming and moisture reduction had no significant effect on mineralization (Bai et al, 2013: Table 1), indicating even in dried samples, pools of inorganic-N may change over time. To remedy this, the authors could have compared their ammonium concentrations to similar studies from this region; however, this was not included in the results/discussion.*

The authors are aware of this particular problem of $NH_3$ volatilization. The analyses have been made as soon as possible after the field campaign. A direct analysis was not possible, due to missing infrastructure. Freezing the samples would have been the best solution, but considering the difficulty of organizing the campaign in a place where the minimum material was installed, we could not afford to bring a freezer.
Some authors have also published results of ammonium concentrations measured in soils that were dried in ambient air (Bai et al., 2010, Dick et al., 2006, Cassity-Duffrey et al., 2015).

Moreover, very few values are available in the literature, it is therefore very difficult to compare. The results presented in section 3.3 are consistent with Massad et al. (2010) which may provide a certain confidence in analysis. Dick et al. (2006) have also found $NH_4^+$ concentrations between 2 and 8 $mgN.kg^{-1}$ in Senegalese soils, which is very close from our results. Vanlauwe et al. (2002) have found values between 0.8 and 1.4 $mgN.kg^{-1}$ in West African moist savanna soils (in Togo and Nigeria).

**Line 255**, new references have been added in the text:
Some authors have also published results of ammonium concentrations measured in soils that were dried in ambient air (Bai et al., 2010, Dick et al., 2006, Cassity-Duffrey et al., 2015).

**Line 360**, the following references have been added:
Dick et al. (2006) have found $NH_4^+$ concentrations between 2 and 8 $mgN.kg^{-1}$ in Senegalese soils, which is very close from our results. Vanlauwe et al. (2002) have found values between 0.8 and 1.4 $mgN.kg^{-1}$ in West African moist savanna soils (in Togo and Nigeria)..

*In regards to the tables and figures, the authors should strongly consider merging Figures 2-5. As they currently sit, there is a large amount of redundancy.*

As reviewer #1 asked to add NO and $NH_3$ concentrations in the figures, we chose not to merge figures 2 to 5, it would have been impossible to read.

**References**

[revised manuscript text omitted]

---

## Author Response (AR2)

Dear Co-editor,

Thanks for this review, which offers us the opportunity to improve and clarify our manuscript. We have corrected the latest version of the manuscript **of 12th June 2018** and added some clarifications (Line numbers in this document refer to the previously reviewed version of the manuscript submitted on 12th June 2018). All corrections are commented below and included in the new version of the manuscript.

```
This review is from the Co-Editor.

This paper describes measurements of NH3 and NO fluxes in Benin, West Africa and
in June and July, 2016. The article has received one round of reviews. My
primary concern is that the new version has not adequately addressed concerns
around the NH3 flux measurements. Data of soil fluxes in West Africa are rare
and I hope more information can be provided on the measurements so the paper can
be published. I have provided other comments as well.

1/The application of experimental results from Vaittinen et al. (2013) is not
yet convincing. The value for PFA of 13.9e12 molecules/cm2 was determined under
constant conditions that do not represent the field sites: 10% sea-level
atmospheric pressure, a constant temperature of 295K, constant humidity, and
ambient NH3 of ~8 ppm (not 8 ppb). Vaittinen et al. (2013) themselves say the
reported uptake values are minimums, rather than absolute quantities. Diurnal
and day-to-day changes in these conditions could drive significant variability
in air-chamber exchange, despite the low standard deviations reported in
Vaittinen et al. (2013). If the chambers were frequently washed, then maybe
uptake is the dominant direction of the air-chamber exchange; however, a key
conclusion is that soils are largely an NH3 sink. For this reason, additional
work is need to verify that the reported fluxes are between the air and soil
rather than the air and chamber.
```

**Authors' response :**

The chambers were not exactly "washed", but rather wiped with dry clean paper cloth. This clarification was added to line 173.

"The Teflon chamber was cleaned with a dry clean paper cloth at the beginning of each day of measurement, and during the day when the deposition of sand could potentially interfere with the measurements"

We propose to modify line 183 with the following text:
"Moreover, experimental tests with and without the Teflon chamber attached to the analyzer were made in ambient air to verify that deposition on the walls of the Teflon chamber is negligible. The results of this experiment are reported in Appendix A."

We propose to add the following text in Appendix A, line 576:

"We ran a laboratory experiment to verify that deposition on the walls of the Teflon chamber is negligible.
Ambient air concentrations were measured by the analyzer, inside the room where the analyzer and the chamber were placed. Measurements of $NH_3$ concentrations were made in ambient air with and without the Teflon chamber attached to the analyzer. The Teflon chamber was placed on a Teflon frame, and they were sealed together with Teflon tape. Measurements of $NH_3$ concentrations with the Teflon chamber attached to the analyzer were followed by measurements without the chamber 30 to 60 minutes later. The two sets of measurements were made under similar conditions of temperature and humidity. Average values of $NH_3$ concentrations were calculated for 10 to 30

minutes before and after connecting the chamber. Average $NH_3$ concentrations during this time interval varied between 8 and 36 ppb, with a variation between 1.5 to 13% around the mean. The lowest $NH_3$ concentrations correspond to air samples previously passed through charcoal and desiccant cartridges (NO and $NO_2$ zero air). Measured $NH_3$ concentrations are reported in Table A1, along with temperature, humidity and the ratio between average concentration with and without the Teflon chamber attached to the analyzer..

This test was made at different times of the day on different days: air humidity varied between 46 and 54%, temperature varied between 25 to 29°C, while pressure varied between 1006 and 1008 hPa (not reported).

Results show negligible variation between concentrations of air reaching the analyzer via the chamber or going directly to the analyzer.

| Temperature (°C) | Humidity (%) | [NH₃] TC | [NH₃] D | Ratio [NH₃]TC / [NH₃]D |
|---|---|---|---|---|
| 25 | 54 | 26.4±0.4 | 25.9±0.5 | 1.02 |
| 27 | 49 | 9.0±0.8 | 8.8±1.2 | 1.12 |
| 29 | 46 | 35.2±2.4 | 36.0±1.4 | 0.98 |
| 28 | 46 | 25.6±0.7 | 24.0±1.0 | 1.07 |
| 27 | 45 | 26.2±0.9 | 27.4±0.9 | 0.96 |
| 28 | 46 | 23.4±0.3 | 24.1±0.4 | 0.97 |
| 26 | 50 | 19.2±0.6 | 18.4±0.8 | 1.04 |

Table A1: Measurements of $NH_3$ concentrations (ppb) through the chamber (TC) or directly (D) to the analyzer. "
* * *
**2/**Referee 1 makes a valid point in comment 2 (below):

"The authors use the measured NO and NH3 fluxes for a stepwise linear multiple regression analysis, upscaling to country-wide soil fluxes, and comparison with soil emission estimates from the GEOSChem model. These analyses give valuable information on the importance of soil NO and NH3 exchange and our current knowledge about them. However, while the authors state that a process understanding of the NO and NH3 fluxes is not within the scope of the presented study, in my opinion it is important to understand the underlying processes of the measured fluxes. For example, the estimated emissions from soil characteristics only poorly agree with the measured fluxes in some parts, which indicates that a more detailed process understanding is necessary."

However, the authors have made no substantive changes to the manuscript in response. Or, if they have, they have not indicated this in the formal response. Please clarify.

**Authors' response :**

As mentioned in the previous authors' response of 12[th] June 2018, the authors agree that a better understanding of the underlying processes of NO and $NH_3$ exchanges is important. As explained in lines 420-425 our experiment set-up was chosen to give an estimate of soil fluxes at a large ecosystem scale, rather than reproducing the relationships between soil fluxes and meteorological variables, like soil temperature and soil moisture. Our experiment does not show the details of microbial and physical processes driving soil fluxes at a single point because measurements are

done at different locations every day, but aims to estimate the spatial variability of fluxes at the ecosystem scale.

We have re-written the two paragraphs starting from line 415 to clarify this:

« Due to this non linear character of the NO fluxes, no direct correlation was found between NO fluxes and environmental variables such as soil moisture or soil temperature taken individually. Moreover, soil temperature and soil moisture were not measured on the same soil parcel where the soil fluxes where measured, and the location of the soil flux measurements was not kept constant even for the same land cover type on the same measurement day. This measurement protocol was designed to give an estimate of soil fluxes at a larger ecosystem scale, rather than reproducing the relationships between soil fluxes and meteorological variables, like soil temperature and soil moisture.

A multiple linear regression analysis was performed between daily mean NO fluxes and the following variables: wind speed, soil temperature at 5 cm, soil moisture at 5 cm, soil heat flux, upward longwave radiation and downward shortwave radiation. This regression gives $R^2$=0.49 (p-value=0.004), indicating a weak but existing relationship between those variables and NO soil emissions, while the regression was weak between NO fluxes and each individual variable. This correlation shows the influence of these environmental variables considered collectively on NO fluxes, highlighting the underlying mechanisms responsible for NO release to the atmosphere. Our experiment does not show the details of microbial and physical processes driving soil fluxes at a single point, because measurements are done at different locations every day, but aims to estimate the spatial variability of fluxes at the ecosystem scale.”

Other studies have been made to investigate the processes behind soil fluxes, see for example lines 45-48 in the Introduction.

Moreover, the comparison of measured and simulated NO soil emissions made in this study is done against GEOS-Chem which includes a process-based model for NO (Hudman et al., 2012). A better process understanding could generally be useful to improve soil emissions models, including the already advanced process-based model by Hudman et al., 2012 used in this study, but other issues should be considered, as the model uncertainty in reproducing weather and soil conditions, as well as the inevitable simplification of the actual highly variable land and vegetation cover introduced by the use of grid cell land cover and vegetation type in the model simulation.
We have modified lines 485-489 to better explain this concept :
« Simulated NO emissions are often higher than those recorded over the grassland areas, however, simulated NO emissions are often within the error bars of measurements (Fig. 6). The model uses land cover and vegetation types to simulate the highly variable land and vegetation cover of the observation site, for this reason we do not expect the model to reproduce the site-to-site variability of the measured soil fluxes, but to at least reproduce their average magnitude and behaviour. It appears that when the model is able to reproduce the length and the intensity of the rain events, NO emissions are especially well simulated, e.g. the model is able to reproduce the longest rain period (from 20 to 26 July 2016) and the decrease of emissions at the end of the measurement campaign. »
* * *
**3/**Rearding Referee 1's comment:

"The assumption that the concentration in the chamber is equal to the
concentration leaving the chamber to the analyzer is questionable. Due to the
low flow rate required for the practical use of the closed-dynamic chamber
technique, the residence time within the chamber is substantial (17-18 min). As
no active mixing (e.g. with fan) is used, the chamber geometry in relation to
the positioning of the ambient air inlet and sample outlet is of importance."

Please modify the text so the reader does not need to read another paper to
understand the fundamentals of the experiment. Consider this comment broadly and
add a brief overview of all relevant information from Delon et al. (2017). You
may refer readers to Delon et al. (2017) for additional detail, but provide the
basics here.

**Authors' response:**
We have added the details of the chamber geometry at line 170:
"The external volume of the chamber was 40 cm × 20 cm × 20 cm. The useful volume was 18 ×38 ×18 cm$^3$, due to the thickness of the Teflon walls".
The structure of the chamber and the positions of the inlet was already included in the latest manuscript, at line 170.
* * *
**4/**For the new text: "This is different from Delon et al., 2017, but fluxes were
anyway superior to this value." Change the word superior to "greater than."

**Authors' response**
We thank the co-editor for this correction. That sentence was only used in the response to the referee and it is not included in the manuscript.
* * *
**5/**Regarding Referee 1's comment (a concern also raised by Referee 2):

"This assumption seems brave if it was not tested with a set of test experiments.
Although the microbial activity is reduced due to the dry conditions, there is a
chance that NH3 volatilizes with the drying of the soil sample material."

I see you have added a reference to this phenomenon, but discussion for how this
relates qualitatively and quanitaviely to your results is still needed.

**Authors' response:**
The paragraph starting from line 247 was re-written to fully discuss this issue.

"Soil samples were collected with a cylinder of known volume (290 cm$^3$) during the measurement campaign to analyze the biogeochemical characteristics of the site. Soil samples (0-5 cm) were taken for each land cover type where NO and NH$_3$ fluxes were measured. Fifteen samples were collected at the four different land cover types, three to four times during the campaign.

Samples were dried in ambient conditions (mean day-time temperature is approximately 26 °C, Kalthoff et al., 2017), and stored in the dark. After drying, the weight of the samples was measured to determine the bulk density ($d_a$ = dry soil mass / total volume), which was found to be 1.24 ±0.14g cm$^{-3}$. Assuming a density of soil particles ($d_r$) of 2.6 g cm$^{-3}$ , the Water Filled Pore Space (WFPS) is calculated with the following equation:

WFPS = SM/(1-d$_a$/d$_r$)                .                                                                                (5)
where SM is the soil moisture in %.

Soil samples were analyzed for the determination of texture, ammonium concentrations [NH$_4^+$], C/N ratio, total C, total N, and pH at the GALYS Laboratoire (http://www.galys-laboratoire.fr, NF EN ISO/CEI 17025: 2005). The analyses were performed two months after sampling. We assume that the ammonium content in litter or soils is not modified by volatilization or chemical transformation during transport and storage, because of the very low soil moisture level in samples. Indeed, when collected, WFPS of the samples ranged between 6 and 14% (mean=8.5±3.5), and soil temperature between 35 and 38 °C (data obtained from the databases described in Brooks et al., 2017). Bai et al. (2013, and references therein) have found that significant changes in nitrification and net mineralization (influencing the ammonium content) may occur when soil temperature is raises up to 35°C (the optimum for nitrification), for optimal soil moisture conditions ( WFPS =20%, Oswald et al., 2013). In the present study, soil temperatures when sampling were equal or above the optimum, and WFPS was below the optimum, reducing the nitrification efficiency and the change in ammonium content. Past studies have demonstrated that microbial activity and mineralization processes are inhibited in low soil moisture conditions, even when soil temperature is high (Bai et al., 2013, and references therein). Several authors have published results of ammonium concentrations measured in soils dried in ambient air. For example, Dick et al., (2006) collected top soil after the wet season in two sites in Senegal. The authors state that their soils were considered dry when collected and were air-dried in the mid-day sun immediately after collection. The protocol used in our study is identical. Other studies (Bai et al., 2010, Cassity-Duffrey et al., 2015, Vanlauwe et al., 2002) also published ammonium measurements made on air-dried soils from seasonally dry climates with comparable textures to the soil in Savé."
* * *
**6/**New text: "Daily means of NO concentration vary from 1.28 to 5.40 ppb for all sites." In Fig. 2 (and 4), NO concentrations appear be equal 7-9 ppb on multiple days.

**Authors' response:**
An error was found in the legend of the figures 2 to 5. The symbols for NO and NH$_3$ concentrations were inverted. This has been corrected. The NO concentration range is correct in the text.
* * *
**7/**As far as I can see, Figs. 2 and 4 are identical. Is this an error?

**Authors' response:**
Yes that is an error. The authors apologize for this error and the wrong figure has been replaced by the correct one for the bare soil site (Figure 2). All legends have been corrected.
* * *
**8/**New text: "Average NH3 concentration is 6.28 ± 3.90 ppb for bare soils." Over bare soil, it appears the NH3 concentration never exceeds ~3 ppb.

**Authors' response:**
As mentioned above the legends for NH$_3$ and NO concentrations were inverted, and figure 2 for bare soils was not the right figure. This was corrected. The concentration ranges given in the text are correct.
* * *
**9/**Regarding Referee 2's comment:

"While I am sure location and resources had much to do with this, air-drying may result in large changes to ammonium concentrations. Additionally, significant changes in the amounts of ammonium can take place over prolonged storage at room temperature, even if soils are dried. It seems that the authors are aware of this issue and attempted to justify their method by citing a meta-analysis of warming experiments on N-cycle activity. (Bai et al. 2013). However, this meta-analysis found that warming and moisture reduction had no significant effect on mineralization (Bai et al, 2013: Table 1), indicating even in dried samples, pools of inorganic-N may change over time. To remedy this, the authors could have compared their ammonium concentrations to similar studies from this region; however, this was not included in the results/discussion."

Please expand the new discussion, which at just two sentences (line 255 and line 360), is insufficient.

**Authors' response:**
Please see authors' response number 5 above, in this document. The paragraph starting from line 247 was re-written to fully discuss this issue.
Some new references were added as asked by the reviewers, however measurements of ammonium content from the same region of this study are apparently not available in the literature.

The discussion on inorganic pool changes with temperature and moisture was included in the newly rewritten paragraph:
* * *
**10/**Other comments:

All plots are of poor visual quality, can the authors remake the plots with a program capable of higher resolution graphics and that allow the authors to improve the readability of the plots?

**Authors' response:**
Figures are provided as pdf for better resolution.
* * *
**11/**Abstract: I don't follow the logic of the first sentence.

**Authors' response:**
The first sentence was modified to:
"Biogenic fluxes from soil at a local and regional scale are crucial to study air pollution and climate. Here we present field measurements of soil fluxes of nitric oxide (NO) and ammonia ($NH_3$) observed over four different land cover types, i.e. bare soil, grassland, maize field and forest, at an inland rural site in Benin, West Africa, during the DACCIWA field campaign in June and July 2016. At the regional scale, urbanization and a massive growth in population in West Africa has been causing a strong increase in anthropogenic emissions. Anthropogenic pollutants are transported inland and northward from the mega cities located on the coast, where the reaction with biogenic emissions may lead to enhanced ozone production outside urban areas, as well as

secondary organic aerosol formation, with detrimental effects on humans, animals, natural vegetation and crops."
* * *
**12/**Line 20: change aerosols to aerosol.
**Authors' response:**
Correction done
* * *
**13/**Abstract should be a single paragraph.

**Authors' response:**
The abstract was rewritten as a single paragraph.

Small errors have been corrected:
Line 156: "reference $NO_2$ air" becomes "reference NO air".
Line 183: "chemical" becomes "photochemical"
Line 188: "increase" becomes "variation"

[revised manuscript text omitted]

---

## Author Response (AR3)

*Dear referee,*

*Thanks for these constructive comments. We have corrected the latest version of the manuscript of 12[th] September 2018. Referee comments are followed by author's responses in italic. We have tried to better characterize the chamber measurements and highlight the uncertainty linked to the sticky character of NH3 and its possible interference with Teflon and particulate matter in air or on the chamber walls.*

The manuscript 'Measurements of nitric oxide and ammonia soil fluxes from a wet savanna ecosystem site in West Africa during the DACCIWA field campaign' reports measurements of nitric oxide (NO) and ammonia (NH3) soil fluxes from four different types of land cover in rural Benin, West Africa in June and July of 2016. The paper is appropriate for ACP and of interest to the journal's readers. It is well written and readable though too many important details of the measurements are missing or in the Delon et al. 2017. I found myself having to refer to Delon et al. 2017 too frequently to try to understand the experiment presented here and making assumptions that the information given in Delon et al. 2017 is applicable to this study.

I am, however, concerned with the lack of characterization of the chamber with respect to the NH3 measurements. NH3 is sticky gas-phase species that both adsorbs and desorbs from surfaces depending on environmental conditions making it difficult to quantitatively sample. As such, NH3 measurements need detailed characterization to assess the quality and uncertainty of the data. As discussed below this manuscript is lacking in these details with respect to the NH3 measurements. In the case of NH3, additional evidence or laboratory work is necessary to ensure that the observations reflect interaction between the air and the soil and not the air and the chamber walls.
* * *
Lines 140-143 – Here it is stated that a Thermoscientific 17i (ThermoFischer Scientific, MA, USA) to measure the concentration of NO and NH3 through chemiluminescence. However, this instrument does not directly measure NH3 via chemiluminescence. Rather it measures NO, NOx by converting NO2 to NO via a molybdenum converter, and N(total) by converting NO2 and NH3 to NO on a stainless steel converter. NH3 is derived via the following equation NH3=N(total)-NOx. A better instrument introduction would improve the manuscript and help the reader understand later details, such as why the instrument was calibrated with NO2.

> *Line 140, the paragraph has been changed into:*

*"The technique used to measure NO and NH$_3$ soil fluxes makes use of a Thermoscientific 17i (ThermoFischer Scientific, MA, USA). This analyzer uses a chemiluminescence detector for NO. The air sample enters the reaction chamber and reacts with the O3 generated by an internal generator. This reaction produces a luminescent radiation directly proportional to the NO concentration. The air sample is sequentially drawn through a molybdenum converter heated to 325°C which measures NOx (NO+NO$_2$) by converting NO$_2$ to NO, and a stainless steel converter heated to 750°C which measures Ntotal (NH$_3$+NOx) by converting NH$_3$ and NO$_2$ to NO. The detector hence measures rNO, then r(NO + αNO2), and finally r(NO + βNO2 + γNH3), where r is the NO detection efficiency, α and β are the NO$_2$ conversion efficiency of the molybdenum and stainless steel converters and γ is the NH$_3$ conversion efficiency of the stainless steel converter. The efficiencies are determined by the calibration*

*procedure. NH₃ concentration is therefore calculated from Ntotal – NOx. The closed dynamic chamber technique is used to calculate fluxes. The details of this technique are fully described in Delon et al. (2017).*
* * *
Lines 151 – 164 ¬- In this paragraph the calibration of the analyzer is discussed. It is good to see the authors calibrate a commercial instrument. However, the details of the calibration need further explanation. As I understand the instrument was calibrated with a NO standard only twice, pre and post-mission. The manuscript should state how these two calibrations compare. The NO2 converter efficiency and the NH3 convert efficiency were each only evaluated once post-mission. This is a weak point in this study. First, the conversion efficiencies of each should be stated. Second, the authors need to show some evidence that the conversion efficiencies do not drift significantly over time because a drift in either could significantly affect the NH3 derived from the instrument.

*Paragraph beginning line 151 was modified as follows:*

*"The calibration of the NO sensor of the 17i analyzer was made before and after the campaign, with a reference NO air mixture, i.e. NO in $N_2$ diluted with zero air. The NO detection efficiency variation was 8% between the two calibrations (from 1.040 to 0.962). Two post-campaign calibrations were made: a first one to validate the efficiency of the $NO_2$ converter using a reference dilution of $NO_2$ in zero air, and a second one to validate the efficiency of the $NH_3$ converter with a $NH_3/N_2$ mixture diluted in pure air (Alphagaz 1, Airliquide). No change in the $NO_2$ conversion efficiency was necessary, and the $NH_3$ conversion efficiency variation was 3% (from 0.963 to 0.995). No drift in the conversion efficiencies was observed over time, as from the first calibration when the analyzer was new until the post campaign calibration, changes never exceeded ±3%. "*
* * *
The authors state in line 163 that 'Multipoint (at least 4 points) calibrations between 50 to 250 ppb were done to ensure the linearity of the response, obtaining regression coefficients over 0.9993 for both NO and NO2.' Does this mean that the multipoint calibrations were done only with the NO and NO2 standards and not the NH3 standard? The ppm level standards were diluted to the 50 to 250 ppb for the multipoint calibrations. What is the uncertainty in the dilution procedure? Were these calibrations performed with the 4m Teflon tube that connected the chamber to the instrument in place? Understanding the effect of this line on NH3 could be important to the NH3 flux determination.

*The following sentence was added line 162, to complete the information on $NH_3$ multipoint calibration*

*"The dilution uncertainty was 10% for NO, 11% for $NO_2$ and 13% for $NH_3$ (see Appendix A for more detail). A multipoint calibration was done for $NH_3$, between 30 and 200 ppb and the regression coefficient was 0.997. The linearity of the response for low concentrations is ensured by the response to zero air calibration with a $R^2=0.997$."*

*Appendix A*

*The dilution uncertainty is calculated based on the uncertainties of standard concentration, standard flow and dilution flow. The uncertainty of standard concentration is 5% for NO and $NO_2$, 2% for $NH_3$. The maximum uncertainty of dilution flow is 1% of the plain scale (10 $L.min^{-1}$) for the three standards divided by the flow used in the diluter (3.2 $L.min^{-1}$ maximum), which gives 0.1/3.2=3.1%. The uncertainty of standard flow is 1% of the plain scale standard flow (50 $mL.min^{-1}$) divided by the standard flow used to obtain the needed concentration (50 ppb for NO or 30 ppb for $NH_3$).*

*Standard flow=(needed concentration/standard concentration)\*dilution flow.*

***For NO***, *dilution flow=3.2 $L.min^{-1}$, needed concentration=50 ppb, standard concentration=8.73 ppm, standard flow=18.4 $mL.min^{-1}$. Uncertainty of the standard flow = 1%\*50/18.4=2.71%. Total uncertainty is therefore 5%+3.1%+2.7%=**10.8%.***

***For $NO_2$***, *dilution flow=3.2 $L.min^{-1}$, needed concentration=50 ppb, standard concentration=9.28 ppm, standard flow=17.2 $mL.min^{-1}$. Uncertainty of the standard flow = 1%\*50/17.2=2.9%. Total uncertainty is therefore 5%+3.1%+2.9=**11%***

***For NH3***, *dilution flow=3.2 $L.min^{-1}$, needed concentration=30 ppb, standard concentration=14.78 ppm, standard flow=6.5 $mL.min^{-1}$. Uncertainty of the standard flow =1%\*50/6.5=7.7%. Total uncertainty is therefore 2%+3.1%+7.7%=**12.8%.***
* * *
Looking through figures 2-5 the measured ambient levels of both NO and NH3 are much lower than 50 ppb. Especially in the case of NH3, which is difficult to sample quantitatively, the authors need to justify their assumption that the instrument response for both is linear below 50 ppb to the ambient levels measured.

*The justification was added line 162 with the preceding comment (The linearity of the response for low concentrations is ensured by the response to zero air calibration with $R^2$=0.997).*
* * *
Line 164 states the global precision of the analyzer is 0.4 ppb. Is this experimentally determined or just the manufacturer's specification?

*0.4 ppb is the manufacturer's specification for a 0-500 ppbv range. This was added line 164*
* * *
Lines 166 – 192 – The description of the chamber is lacking in important details. Here the manuscript would benefit with a good schematic showing the locations of the air inlet and the outlet to the chamber, i.e. the geometry and layout of the chamber. The residence time of air in the chamber is given as 20 minutes. Yet the chamber is maintained in place for only 10 minutes, i.e., less than 1 turnover of air in the chamber. Without a fan and depending on the geometry of the inlet and outlet, it seems possible that air in the chamber could be layered such that the mixing ratio of NO or NH3 going to the instrument is equal to that in the chamber. How does this affect the flux determination? Why do the authors feel this is not an issue?

*Line 169, we added a reference to a picture of the chamber displayed in Appendix B, and the picture was added line 630 in Appendix B.*

*We have recently made some tests in the field to verify if mixing would have an effect on the concentration increase in the chamber. In Appendix B, the following text was added:*

*"To verify if mixing the air in the chamber by a fan would change the shape of the increase in concentration or the concentration itself in the chamber, a test was made with a syringe simulating the action of a fan (i.e. we have mixed the air inside the chamber by sucking and releasing the same air with a syringe through the small vent, while letting outside air entering the chamber by the vent as usual to ensure pressure equilibrium between outside and inside air). The comparison between a flux measurement with and without mixing gives similar slopes of the concentration increase (or decrease)."*

*As the flux is calculated from the first minutes after placing the chamber on the ground, the flux calculation would not change even if one turnover of air in the chamber is not achieved.*
* * *
The manuscript cites Vaittinen et al. 2013 to support their claim that the effect of the Teflon walls on NH3 is negligible. However, this is not thoroughly convincing because the experiments performed in this study were with NH3 mixing ratios in the hundreds of ppb to low ppm range, not applicable to the atmospheric levels seen here. Furthermore this study was performed in a much more controlled environment and thus not directly comparable. Similarly, the experiment in Appendix A is promising though the authors should better relate the laboratory conditions to those experience in the field. One aspect that does not seem to be covered by the Appendix A experiments is the temperature of the Teflon itself. When sitting in direct sunlight, the temperature of the Teflon surface could differ from the air temperature and affect the adsorption or desorption of NH3. Additionally, the 4m Teflon tube from the chamber to the instrument could exhibit surface affects that differ from the chamber walls.

*The paragraph beginning line 182 was modified:*

*"The calculation of the fluxes are based on the closed dynamic chamber technique, with the following assumptions: the concentration in the chamber is equal to the concentration leaving the chamber to the analyzer, and potential deposition onto the Teflon walls of the chamber is assessed but considered as negligible. Vaittinen et al. (2013, and references therein) have demonstrated that the adsorption of ammonia on Teflon is negligible; However, the high $NH_3$ mixing ratios and the controlled conditions in Vaittinen experiment do not correspond to our field conditions. Therefore, experimental tests with and without the Teflon chamber attached to the analyzer were made in ambient air to verify that deposition on the walls of the Teflon chamber is negligible. These tests have been made in conditions comparable to in situ measurements, i.e. temperature (25 to 29°C) and humidity (46 to 54%), as well as $NH_3$ concentrations (8 to 35 ppb) close to the ones encountered in the field. They show that the concentrations measured with and without the chamber are equivalent. The results of this experiment are reported in Appendix B. Moreover, the temperatures of the chamber Teflon walls and Teflon tube have been measured in direct sunlight and the difference with air*

*temperature is small (<1°C). We therefore assume that the Teflon walls and tube heating is small and does not affect the NH$_3$ and NO concentration measurements in the chamber. The results of this experiment are reported in Appendix D. All the details of the calculation are given in Delon et al. (2017). "*

*Line 194, the reference to Delon et al. 2017 has been replaced by "The dilution effect due to mixing of outside air in the chamber was evaluated based on our set up in which Q/V=8.13×10$^{-4}$ s$^{-1}$."*

*Line 630:*

*Appendix D*

| Location of the temperature measurement | Temperature (°C) |
|---|---|
| Air | 32.7 |
| Soil | 34.2 |
| Chamber: outside wall | 33.1 |
| Chamber: inside wall | 33.3 |
| Chamber: outside top | 33.8 |
| Tube: outside close to the chamber | 30.5 |
| Tube : outside close to the analyzer | 30.3 |
| Tube: inside | 30.9 |

*Table D1 : Temperature measured on the Teflon chamber and on the Teflon tube. These measurements have been made after the field campaign in direct sunlight at 3:30 PM. Measurements were made with a calibrated thermometer  HI 98509 with stainless steel probe (-50 → +150 °C).*

The effect of cleaning the chamber daily should also be investigated. Though it appears that no chemicals were used in the clean, does the daily wiping remove NH3 or add NH3 from the chamber surface? If NH3 is removed, then, are observations of deposition really reflecting NH3 deposition on the chamber wall not the soil? Perspiration is a significant source of NH3. In the act of clean could the walls adsorb NH3 and then come off during a sampling period. These are hypothetical but could affect the flux determination, nevertheless.

*Some more experiments were done to check a possible interaction of cleaning paper and Teflon walls, with different types of paper used (A, B and C). These tests were made in laboratory conditions with mean temperature around 24°C in the room.*

*Line 171, the following sentence was added:*

*"Laboratory tests using different papers for cleaning are displayed in Appendix C. According to these results, no clear tendency for potential adsorption or desorption of NH$_3$ arises but these tests may be useful to warn for a potential pollution inside the chamber due to cleaning which would interfere with low fluxes."*

*Appendix C*

*NH$_3$ concentration was recorded continuously inside the Teflon chamber (placed on a Teflon material as in the tests summarized in Appendix B), and the chamber was cleaned successively with three different papers, referred to as A, B and C. The concentration was recorded at least for 30 minutes*

*between every cleaning. Table C1 summarizes the averaged concentrations (and standard deviations) for every period. Results show a variation of concentration when different papers are used, but this variation is not reproducible and is difficult to differentiate from a natural variation of the NH₃ concentration in the room. As a matter of fact, the effect of cleaning on NH₃ adsorption or desorption is not clear, but questions about the potential pollution of the chamber arise. Results in Table C1 may lead to the conclusion that if the difference in concentration during a flux measurement is inferior to a certain threshold, it is not necessarily a flux from the ground but could be due to an adsorption or desorption of NH₃ by chamber walls due to cleaning. Only low fluxes are concerned. To set ideas down and take an example, fluxes inferior to 0.5 ngN.m-2.s-1 represent 23% of the 350 measured fluxes. If those low fluxes were removed from the database, the resulting average would be slightly larger in magnitude (-1.1 instead of -0.9 ngN.m-2.s-1). As a conclusion, these tests may help to warn the reader that caution must be kept for low NH₃ fluxes because of possible pollution in the chamber.*

| *Paper used for cleaning* | *30 minutes average (standard deviation) in ppb* | *Difference between two successive averages in ppb* |
|---|---|---|
| | *First day of test* | |
| *Before cleaning* | *8.13±0.58* | |
| *A* | *7.43±0.47* | *-0.69* |
| *B* | *8.55±0.79* | *1.12* |
| *C* | *9.73±0.81* | *1.18* |
| *B* | *9.75±0.88* | *0.02* |
| | *Second day of test* | |
| *B* | *16.35±0.92* | |
| *C* | *16.77±0.60* | *0.42* |
| *A* | *17.30±0.76* | *0.53* |
| *A* | *18.94±0.72* | *1.64* |
| *B* | *18.96±0.62* | *0.02* |
| *C* | *18.79±0.79* | *-0.17* |

*Table C1: 30 minutes averaged concentrations in the Teflon chamber after cleaning with different dry papers.*
* * *
Line 225 – More in-depth reasoning should be given as to why interaction with particulate matter is ignored, especially in regards to NH3.

*Line 225, the sentence was modified:*

*"NH₃ measurements have not been corrected from a possible interaction with particulate matter (PM) as PM concentration (not measured at Savé) are supposed to be low because Savé is located in a rural area far from anthropogenic pollution influence. Moreover, the walls of the Teflon chamber are cleaned daily to reduce any interference of NH₃ with PM deposition in the chamber. The effect of PM, even at low PM concentrations, may reduce the measurement accuracy and induce an uncertainty on the detection of the NH₃ flux from soil. This uncertainty has not been assessed quantitatively, but the reader must keep in mind that NH₃ fluxes may be estimated with less accuracy because of the presence of PM, especially for low fluxes."*
* * *
Lines 227 – 235 – Here the manuscript could be improved by actually showing chamber data, i.e., how the measured concentration varies with time. No explanation is given why the R^2 threshold for NO is 0.8 while that for NH3 is 0.4. Is this indicating that there are other influences on the NH3 mixing ratios in the chamber such as wall adsorption or particulate matter interaction? Is there a difference in derived fluxes for NH3 when the R^2 is 0.4-0.6 compared when the R^2 was above 0.8?

*The following text was added line 229:*

*"The variation of $NH_3$ is less stable than for NO because of potential interaction with PM in the chamber. However, 80% of the $R^2$ were superior to 0.6 for $NH_3$, and 100% for NO. Examples of the variation in time of the concentration of $NH_3$ and NO in the chamber are shown in Appendix E for two different soils"*

*Line 278, the following paragraph was added:*

*"Uncertainty of the $NH_3$ flux calculation*

*Despite all precautions to reduce adsorption on the chamber walls and/or interaction with PM in air, (see Appendix B, C, D and E), $R^2$ are less good for $NH_3$ due to potential chemical or physical interaction of material with $NH_3$ (whereas considered as negligible in this study). However, no absolute correction for adsorption can be calculated in field conditions. Teflon remains the more reliable material to measure $NH_3$, as shown in Sauren et al. (1989) who find that Teflon has the lowest adsorption affinity for $NH_3$ (as compared with aluminium, parafine and gold), but a passivation time lag remains for $NH_3$ detection in measurements systems (Yokelson et al., 2003)."*

*Appendix E:*

*NO and $NH_3$ fluxes are calculated from the slope of their concentration increase (or decrease) in the chamber through time. Two examples are given in figures D1 to illustrate the larger instability of $NH_3$ detection compared to NO detection, due to possible interaction of $NH_3$ with chamber walls, particulate matter or humidity in the chamber.*

[Figure]

*Figure E1: NH$_3$ (a,c) and NO concentration (b,d) variation with time (one point every 10 second) inside the chamber on grassland (left) and bare soil (right).*
* * *
Tables – Tables 3, 4, and 5 summarize the soil measurements. There are no comparable tables for the atmospheric observations, e.g., the average NH3 mixing ratio over the various land types (line 468).

*Table 6 was added to summarize fluxes and concentrations for all cover types. References to Table 6 were added in appropriate places in paragraphs 3.4 and 3.5.*
* * *
Figures – It is difficult differentiating the different symbols in Figures 2-5. Consider using color as in Figure 6 to help the reader more easily follow the data.

*Figures 2 to 5 were changed.*
* * *
Minor comments

Line 36 – Oxide does not need to be capitalized.

*Corrected*

Line 166 – It is simpler to say internal volume instead of useful volume as useful is not defined in this context.

*Corrected*

Line 191 – There is no Davidson et al (1990) in the reference list. Please change to appropriate reference. Also, I suggest showing the conversion of the flux to ngN m^-2 s^-1 either here or in an appendix.

*Reference was changed by Davidson et al. (1991) and added in the reference list*

Line 220 – I fail to see how low calculate values of O3 near the soil indicate that O3 deposition is of secondary importance when these calculated values have not been verified.

*This assumption could not be verified by measurements. We have verified that if ozone concentration is 1 ppb ($1e^{-9}$ mol/mol), for an ozone deposition velocity of 0.2 cm/s typical for Benin ecosystem (Adon et al., 2013), the ozone deposition flux would be $8.10e^{-12}$ $mol.m^{-2}.s^{-1}$. The correction for photochemistry is 8% of 4.8 $ngN.m^{-2}.s^{-1}$, i.e. 0.4 $ngN.m^{-2}.s^{-1}$, or $2.9*10^{-11}$ $mol.m^{-2}.s^{-1}$. The quantity of ozone deposited will not change dramatically the quantity of ozone chemically reacting with NO. These calculations were not inserted in the manuscript.*

Line 410 – Here daily means of NO are reported. However, measurements were not taken over the course of a full day. Rather, line 135 states that each location was sampled during the daytime. I'd suggest changing here and elsewhere to 'daytime means'.

*Corrected everywhere daytime values have been used.*

Line 466 – I do not understand what is meant by 'NH3 concentration vary from 0 to 12.46 ppb for all sites'. How can this instrument report 0?

*"nearly" has been added in the sentence*

Line 468-468 – 'and 3.68 ± 2.13 440 ppb for forest.' Appears to be a typo

*Corrected*

[revised manuscript text omitted]

---

## Author Response (AR4)

*Dear referee,*

*Thanks for the constructive comments.*

*We address the reviewer's comments here below, in italics.*

*We have also updated  the latest version of the manuscript of 10[th] December 2018 to address the reviewer's comments.*

This review is focused specifically on the experimental design and interpretation of the NH3 flux measurements during this study. It is well-known that NH3 is susceptible to adsorption and desorption processes that are temperature and humidity-dependent and the authors acknowledge this issue throughout the manuscript. Several lines of evidence are provided to evaluate the potential for artefacts in the flux measurements. Given how rare direct measurements of NH3 flux are, and how challenging they can be to obtain, I support the publication of the data in this manuscript, however I have some suggestions for the authors to slightly reframe their discussion of the results.

On lines 175 – 180, the authors state that calibration linearity at lower concentrations is assured by a 4-point linear calibration between 30 and 200 ppb, and the response to a 'zero air calibration'. However, several studies (e.g. Whitehead et al., 2008, Ellis et al., 2010) have reported that biases related to adsorption and desorption become relatively more important at lower concentrations, so one cannot simply interpret linearity and high concentrations and a clean background as evidence that there are no issues with quantification at lower concentrations. Given that the ambient mixing ratios are mostly below 10 ppb, the authors should be more cautious in their interpretation.

> *Authors response: Line 177, the following text has been added:*

*"The linearity of the response for low concentrations is tested by the response to zero air calibration, giving a $R^2$=0.997. However, at low mixing ratios (typically less than 100 ppb), a non linear increase of the interactions of $NH_3$ with the surface used in the inlet design has to be considered (Ellis et al., 2010, Whitehead et al., 2008). Therefore, an uncertainty in the quantification of low $NH_3$ concentrations has to be taken into account due to surface interactions. The global precision of the analyzer is ±0.4 ppb according to the manufacturer's specification for a 0-500 ppbv range."*
* * *
In Appendix B, the authors present a set of laboratory experiments to evaluate the potential impact of adsorption or desorption from the chamber walls under a narrow range of temperature and RH. I believe that the concentration differences between the through chamber (TC) and direct (D) measurements seen here would be a better way to estimate of the detection limit for fluxes than the 0.4 ppb instrument precision limit that the authors are currently using. For example, one could say that an absolute difference of 1.6 ppb or a relative difference of 12% can arise simply from the effects of adsorption or desorption. In this case, ambient flux measurements that generate such small differences in NH3 should be viewed as not statistically different than zero.

*Authors response: An error has been found in Table B1, line 3 column 4, [NH$_3$]D is 8.0 instead of 8.8.*

*According to Table B1 and Appendix B, an absolute difference of maximum 1.6 ppb or a relative difference of maximum 12% can be due to interactions of NH$_3$ with the surface. Instead of considering these maximum values, we calculated the average difference in concentration between TC and D measurements, which is 0.9 ppb.*

*Line 672, the following text has been added: "The average difference in concentration is 0.9 ppb, and should be considered as the detection limit for fluxes significantly different from zero (i.e. including potential effects of adsorption or desorption). 30% of the concentration differences are below 0.9 ppbv."*

*Line 221, the following text has been added:" According to Appendix B, if the difference in NH$_3$ concentration used to calculate a flux is inferior to 0.9 ppb, the resulting flux may not be distinguished from a potential effect of adsorption or desorption onto the chamber walls. The precision of the analyzing device (analyzer + chamber + tube) may be defined at 0.9 ppb (corresponding to a flux of 0.55 ngN.m$^{-2}$.s$^{-1}$). In that case, low NH$_3$ fluxes comprised between 0.4 and 0.55 ngN.m$^{-2}$.s$^{-1}$ are considered as close to zero but are kept in the average daily flux calculation."*

*Line 266, the following text was added: ". Among the 351 NH$_3$ valid fluxes, 30% are derived from a concentration difference of less than 0.9 ppb."*
* * *
A consequence of this more conservative approach will be that the number of NH3 flux measurements above the threshold for quantification will decrease. The authors expand on this idea in the discussion in Appendix C, indicating that 23% of the fluxes measured were below a value of 0.5 ng N m-2 s-1. The authors go on to speculate about the consequence of excluding these low flux values from the database – the resulting average flux value is larger. I think that the low values should not be excluded – they contain important information about periods where the fluxes are small. Rather the authors should use the results from the experiments in Appendices B and C to derive a more realistic flux detection limit for NH3 and then clearly report that for positive or negative fluxes of smaller magnitude, the flux is statistically indistinguishable from zero. These near-zero values are still meaningful – for example they can be compared to the predictions from the compensation point framework described in Section 2.7, or compared to the much larger fluxes measured over fertilized maize fields in other parts of the world.

*Author response: The definition of a new detection limit has already been discussed in the previous response. We have modified Appendix C and added the following sentence:*

*Line 697: "Positive or negative fluxes inferior to 0.55 ngN.m$^{-2}$.s$^{-1}$ (corresponding to concentration differences less than 0.9 ppb in the flux calculation,as defined in Appendix B) represent 30% of the 350 measured fluxes".*

*In the results and discussion part, the fluxes are daily averaged. It is therefore rather difficult to compare near zero individual fluxes with daily averages.  To include their existence in the discussion part, the following sentence was added:*

*Line 494: " As discussed in Appendices B and C, 30% of individual fluxes used to calculate the daily averages are very low and not distinguishable from adsorption or desorption of $NH_3$ on chamber walls. These very low fluxes are however meaningful and indicate that some periods of near zero measurements must be taken into account to represent the processes of exchanges in these ecosystems."*
* * *
Additional specific comment: The statement on lines 53-55 implies that application of synthetic fertilizer is the largest source of ammonia from agriculture whereas it is actually animal husbandry. The sentence should be modified accordingly.

[revised manuscript text omitted]